# Bmal1 regulates circadian expression of cytochrome P450 3a11 and drug metabolism in mice

Yanke Lin[1,3], Shuai Wang[1,2,3], Ziyue Zhou[1], Lianxia Guo[1], Fangjun Yu[1] & Baojian Wu[1]*

Metabolism is a major defense mechanism of the body against xenobiotic threats. Here we unravel a critical role of Bmal1 for circadian clock-controlled Cyp3a11 expression and xenobiotic metabolism. Bmal1 deficiency decreases the mRNA, protein and microsomal activity of Cyp3a11, and blunts their circadian rhythms in mice. A screen for Cyp3a11 regulators identifies two circadian genes *Dbp* and *Hnf4α* as potential regulatory mediators. Cell-based experiments confirm that Dbp and Hnf4α activate *Cyp3a11* transcription by their binding to a D-box and a DR1 element in the Cyp3a11 promoter, respectively. Bmal1 binds to the P1 distal promoter to regulate Hnf4α transcriptionally. Cellular regulation of Cyp3a11 by Bmal1 is Dbp- and Hnf4α-dependent. Bmal1 deficiency sensitizes mice to toxicities of drugs such as aconitine and triptolide (and blunts circadian toxicity rhythmicities) due to elevated drug exposure. In summary, Bmal1 connects circadian clock and Cyp3a11 metabolism, thereby impacting drug detoxification as a function of daily time.

[1] Research Center for Biopharmaceutics and Pharmacokinetics, College of Pharmacy, Jinan University, 601 Huangpu Avenue West, 510632 Guangzhou, China. [2] Integrated Chinese and Western Medicine Postdoctoral research station, Jinan University, 601 Huangpu Avenue West, Guangzhou, China. [3]These authors contributed equally: Yanke Lin, Shuai Wang. *email: bj.wu@hotmail.com

Circadian rhythms are found in virtually all living things including animals, plants and microbes. In mammals, many aspects of physiology and behaviors (e.g., blood pressure, hormone release, heartbeats and sleep-wake cycles) are subjected to circadian rhythms, allowing adaptation of physiological needs to the time of day in an anticipatory way. Disruption of circadian rhythms has been linked to various health problems such as sleep disorders, obesity, diabetes, and cancers[1]. Circadian rhythms are produced and regulated by circadian clocks that are organized in a hierarchical and interactive manner[2]. The master clock located in the suprachiasmatic nucleus of hypothalamus coordinates all peripheral clocks through neuronal connections and hormonal signals[3,4]. At the molecular level, all circadian clocks are composed of a network of genes/proteins that form several feedback circuits[5]. The main circuit consists of *BMAL1* (brain and muscle ARNT-like 1, *ARNT1*) and *CLOCK* (circadian locomotor output cycles kaput) (or *NPAS2* (neuronal PAS domain protein 2)) as the positive elements, as well as *CRY* (cryptochrome) and *PER* (period) as the negative elements[6]. The BMAL1/CLOCK heterodimer activates the transcription of clock-controlled genes (CCGs) including *CRY/PER*[6]. Once reaching a critical concentration, CRY/PER proteins inhibit the transcriptional activity of BMAL1/CLOCK, thereby downregulating their own expressions[7]. When *CRY/PER* are unable to repress BMAL1/CLOCK, a new cycle of *CRY/PER* (and other CCGs) transcription can start. Additional transcriptional factors such as RORs/REV-ERBs and DBP (albumin site D-binding protein)/E4BP4 (E4 promoter-binding protein) contribute to circadian rhythms via regulation of BMAL1 and PER[8]. In addition to transcriptional regulation, post-transcriptional modifications (e.g., phosphorylation by CKI (casein kinase I) and GSK3 (glycogen synthase kinase-3)) play a role in maintaining the robustness of circadian circuits[9].

Metabolism (biotransformation) is a main mechanism for xenobiotic/drug detoxification in the body[10]. Cytochromes P450 (CYPs) are a major class of enzymes responsible for xenobiotic metabolism[11]. Of all CYP enzymes, CYP3A4 (Cyp3a11 in mice) is perhaps the most important one as it contributes to the metabolism of about 50% of drugs[12]. Many drug-processing genes including *CYPs* present temporal variations in tissue expression[13,14]. Circadian expression of these genes underlies the chronopharmacokinetics, contributing to circadian time-dependent drug efficacy and/or toxicity[15]. For instance, the temporal variations in the hypnotic effects of hexobarbital are attributed to the circadian oxidase activities in rats[16]. Mitoxantrone toxicity closely depends on dosing-time with the least toxicity and the highest drug clearance at 16 h after light onset in mice[17].

Circadian rhythms in xenobiotic metabolism have been recognized[18,19]. A previous study shows that the clock output genes (i.e., the three PAR basic leucine zipper (bZip) family of transcription factors, *Dbp*, *Hlf* (hepatocyte leukemia factor) and *Tef* (thyrotroph embryonic factor)) control diurnal expression of Cyp2b10 via rhythmic regulation of the CAR receptor, a well-known activator of the enzyme[20]. This original study suggests molecular clock-controlled metabolism and detoxification. However, whether and how the clock machinery (and its components such as *Bmal1* and *Clock*) controls drug-metabolizing enzymes remain unknown. One of more important questions awaits investigations: whether and how Cyp3a11/CYP3A4 (the most important enzyme for drug metabolism/detoxification) is controlled by the clock system.

The BMAL1/CLOCK heterodimer essentially is a transcriptional activator that stimulates the transcription of target genes (including *PER* and *CRY*) via binding to the E-box elements present in promoter regions[6,21]. Although in most cases both BMAL1 and CLOCK are required in transactivation of clock-controlled genes, BMAL1 can independently regulate the expression of certain genes such as *Dio2* (type II iodothyronine deiodinase) and *p21*[Waf1/CIP1][22,23]. In addition to controlling circadian rhythms, BMAL1 has regulatory roles in the development of various diseases such as hypertension, diabetes, dyslipidemia and obesity[24,25].

In the present study, we aimed to investigate the role of the core clock component Bmal1 in circadian regulation of Cyp3a11 (playing a leading role in xenobiotic/drug metabolism), and to determine the molecular mechanisms involved. We further clarified the impact of Bmal1 on time-dependent xenobiotic metabolism and detoxification. Overall, to our knowledge, our results defined a previously unreported paradigm for circadian clock-controlled xenobiotic detoxification.

## Results

**Bmal1 ablation blunts the circadian rhythms of Cyp3a11 expression and activity.** Bmal1 knockout mice were generated by deleting a portion of exon5 using the CRISPR/Cas9 technique (Supplementary Fig. 2), and validated by PCR genotyping and expression profiling (Fig. 1a−c and Supplementary Fig. 3). As expected, hepatic and intestinal Cyp3a11 showed a robust circadian expression in wild-type mice (Fig. 1d, e and Supplementary Fig. 4A). However, global deletion of Bmal1 decreased Cyp3a11 expression, and blunted its circadian rhythmicity in both liver and small intestine (Fig. 1d, e and Supplementary Fig. 4A). Bmal1-deficient mice showed reduced metabolic activities toward testosterone and midazolam (two known Cyp3a11 substrates) consistent with the expression changes of Cyp3a11 (Fig. 1d, e). Likewise, Bmal1 knockdown led to downregulation of *Cyp3a11/CYP3A4* in various hepatoma and colon carcinoma cell lines (i.e., Hepa1-6, Hepa-1c1c7, CT26, HepG2, and Caco-2) (Fig. 1f, g). Additionally, overexpression of Bmal1 caused increases in Cyp3a11 mRNA and protein in Hepa1-6 cells (Fig. 1h and Supplementary Fig. 4B). Bmal1 induced the promoter activity of a −2.0 kb *Cyp3a11* reporter in the luciferase reporter assays (Fig. 1i). Taken together, these data supported a critical role for Bmal1 in circadian regulation of Cyp3a11.

**Bmal1 ablation downregulates Dbp and Hnf4α.** No canonical E-box element (a DNA motif for Bmal1 binding and transactivation) was found in the promoter region of mouse Cyp3a11 based on in silico sequence analysis (Genomatix program). Therefore, an indirect rather than direct mechanism was involved in Bmal1 regulation of Cyp3a11. Interestingly, of six known Cyp3a11 regulators, *Dbp* and *Hnf4α* mRNAs were decreased in mouse liver and small intestine as a result of Bmal1 deletion (Fig. 2a, b). In line with the mRNA changes, Bmal1 deficiency caused reductions in Dbp and Hnf4α proteins (Fig. 2c, d and Supplementary Fig. 5). In addition, Bmal1 knockdown resulted in reduced expressions of *Dbp* and *Hnf4α* in both Hepa-1c1c7 and CT-26 cells (Fig. 2e, f). These data suggested potential roles of Dbp and Hnf4α in Bmal1 regulation of Cyp3a11. It was noted that the expressions of the other two PAR bZip genes Tef and Hlf were unaffected by Bmal1 deletion (Supplementary Fig. 6).

**Dbp and Hnf4a mediate Bmal1 regulation of Cyp3a11.** Overexpression of Bmal1 or Dbp or Hnf4α led to an increased level of *Cyp3a11* mRNA in Hepa-1c1c7 cells (Fig. 3a). Luciferase reporter assays revealed a *Cyp3a11* promoter region of −220/+78 bp as a key determinant to Bmal1 responsiveness (Fig. 3b). A non-canonical E-box (−1488/−1482 bp) was found in *Cyp3a11* promoter. However, mutation of this E-box did not alter the effect of Bmal1 on the Cyp3a11 reporter (−2.0 kb) activity (Fig. 3c). ChIP

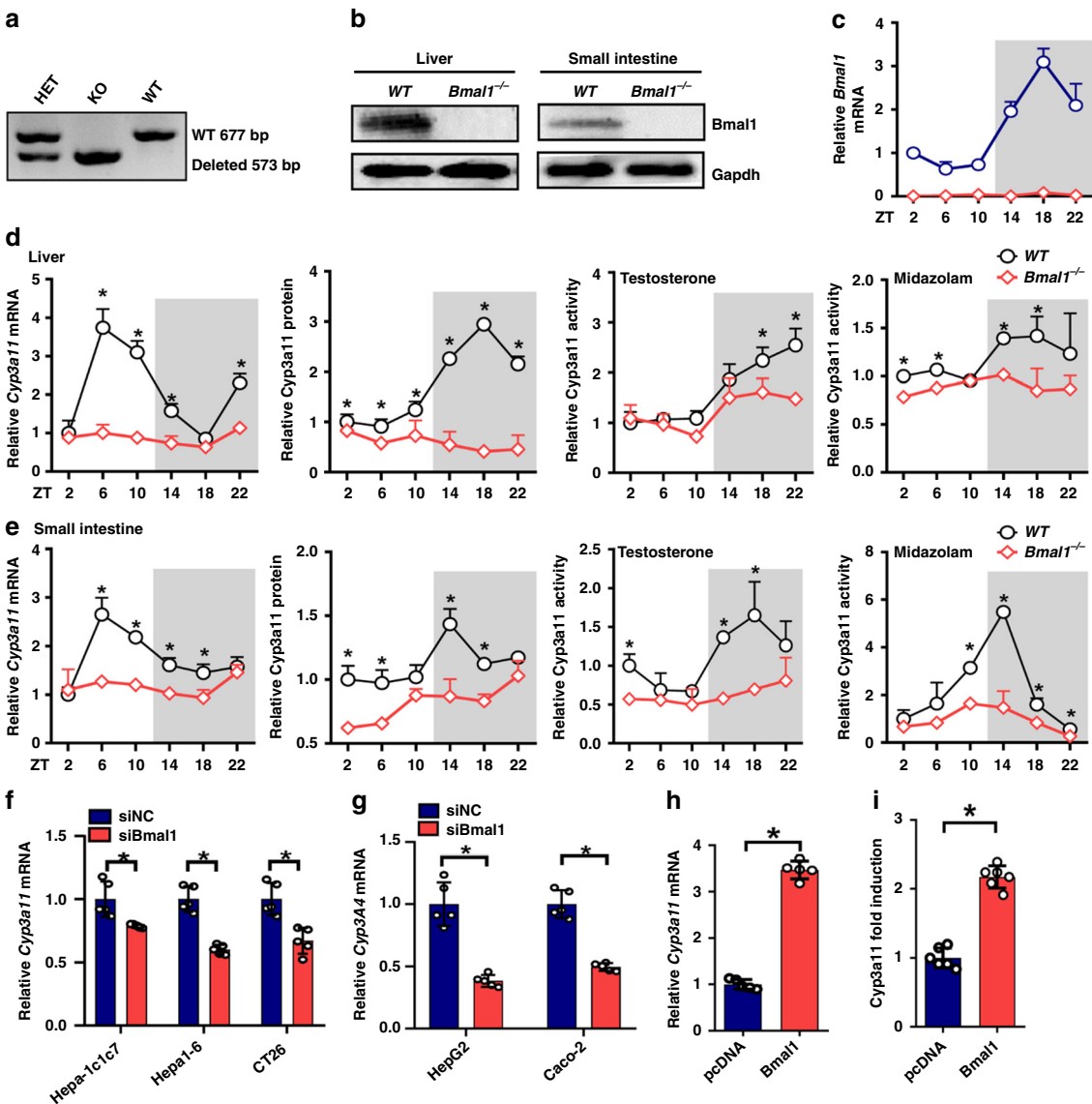

**Fig. 1 Bmal1 ablation blunts the circadian rhythms of Cyp3a11 expression and activity. a** Mouse genotyping result generated from agarose gel electrophoresis. **b** Bmal1 protein expression in the liver and small intestine of wild-type and $Bmal1^{-/-}$ mice. **c** Circadian Bmal1 mRNA profile in the livers of WT and $Bmal1^{-/-}$ mice. Data are mean ± SD ($n = 5$). **d** Circadian mRNA level, protein level and microsomal activity of Cyp3a11 in the livers of WT and $Bmal1^{-/-}$ mice. Data are mean ± SD ($n = 5$). *$P < 0.05$ for two group comparisons at individual time points (post hoc Bonferroni test). **e** Circadian mRNA level, protein level and microsomal activity of Cyp3a11 in the intestines of WT and $Bmal1^{-/-}$ mice. Data are mean ± SD ($n = 5$). *$P < 0.05$ for two group comparisons at individual time points (post hoc Bonferroni test). **f** Cyp3a11 mRNA expression in Hepa1-6, Hepa1c1c7 and CT26 cells transfected with siRNA of Bmal1 or negative control. *$P < 0.05$ ($t$ test). **g** CYP3A4 mRNA expression in HepG2 and Caco-2 cells transfected with siRNA of Bmal1 or negative control. *$P < 0.05$ ($t$ test). **h** Cyp3a11 mRNA expression in Hepa1-6 cells transfected with Bmal1 or pcDNA. *$P < 0.05$ ($t$ test). **i** Luciferase reporter assays with Hepa1-6 cells, showing the effects of Bmal1 on Cyp3a11 transcription. Data are mean ± SD ($n = 6$). *$P < 0.05$ ($t$ test). Unshaded box represents the light period and the shaded box represents the dark (lights off) period

assays showed no recruitment of Bmal1 protein to this "E-box" (Fig. 3d), supporting that the noncanonical E-box was nonfunctional. In-depth sequence analysis of the −2.0 kb promoter identified a DR1 (−123/−110 bp) and a D-box (−45/−36 bp) motif to which Hnf4α and Dbp might bind, respectively (Fig. 3e). Recruitment of Hnf4α to the DR1 motif (−123/−110 bp) of Cyp3a11 was confirmed by a ChIP assay (Fig. 3f). Dbp enhanced the promoter (−2.0 kb) activity of Cyp3a11 in luciferase reporter assays (Fig. 3g). However, this activation effect was lost when the D-box of −45/−36 bp was mutated (Fig. 3g). Similarly, Hnf4α increased the Cyp3a11 reporter activity, and this effect was abolished when the DR1 of −123/−110 bp was mutated (Fig. 3h and Supplementary Fig. 7). By contrast, a mutation of D-box or

DR1 alone attenuated but was unable to abrogate Bmal1-mediated activation of Cyp3a11 reporter (Fig. 3i). A dual mutation of D-box and DR1 eliminated the responsiveness of Cyp3a11 reporter to Bmal1 (Fig. 3i). Knockdown of Dbp or Hnf4α alone (by specific siRNA) caused a reduction in Cyp3a11 mRNA in Hepa-1c1c7 cells (Fig. 3j). The knockdown efficiencies of siDbp and siHnf4α were verified by western blotting (Supplementary Fig. 8). The extent of Cyp3a11 mRNA reduction was more significant when both were knocked-down (Fig. 3j). Moreover, the effect of Bmal1 on Cyp3a11 was attenuated when Dbp and/or Hnf4α were silenced (Fig. 3j). Also, Bmal1 ablation led to reductions in respective recruitments of Hnf4α and Dbp proteins to the DR1 (−123/−110 bp) and D-box (−45/−36 bp) elements

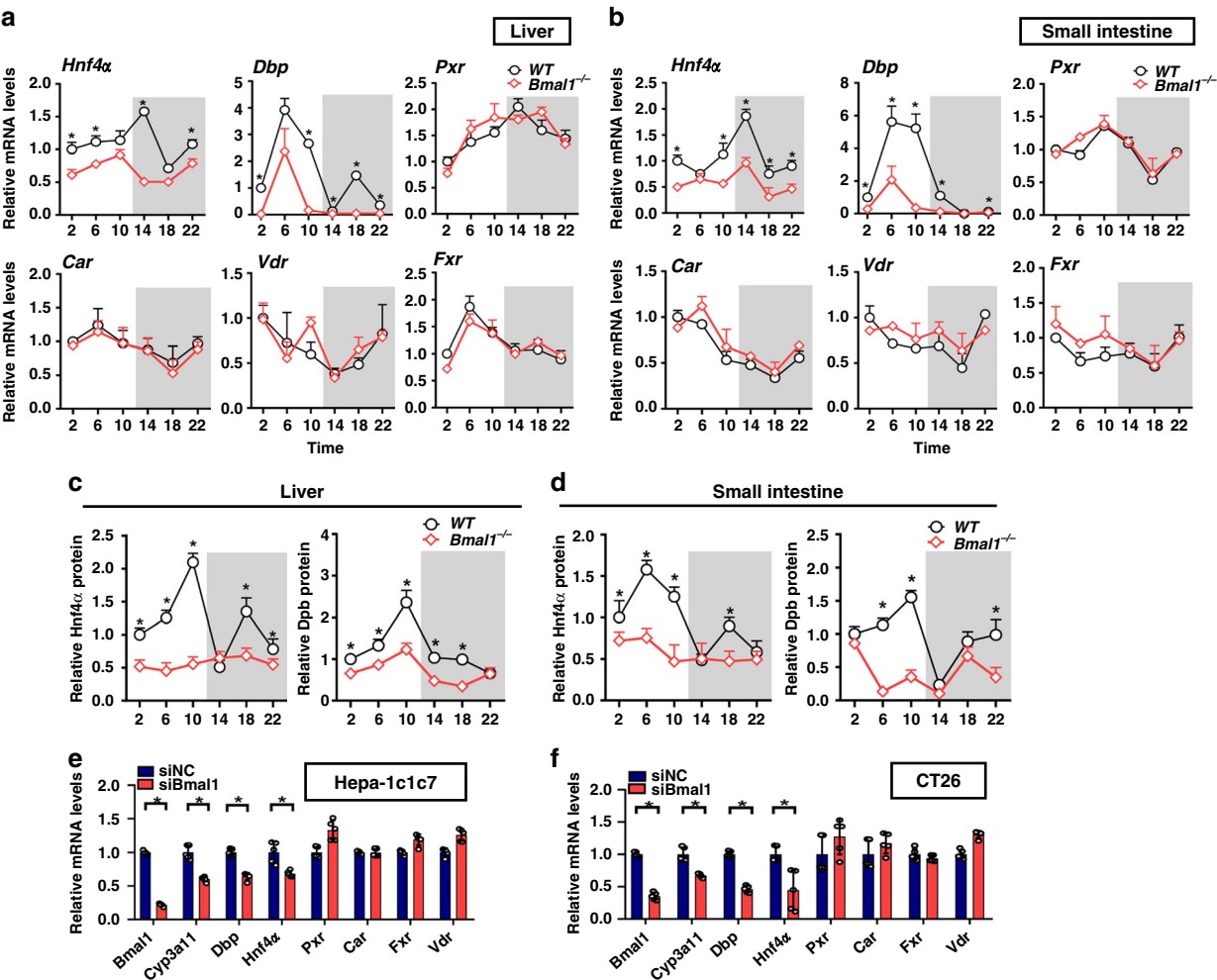

**Fig. 2** Bmal1 ablation downregulates Dbp and Hnf4α. **a** mRNA expressions of six Cyp3a11 regulators in the livers of WT and *Bmal1⁻ᐟ⁻* mice at ZT2, 6, 10, 14, 18 and 22. Data are mean ± SD (*n* = 5). **P* < 0.05 for two group comparisons at individual time points (post hoc Bonferroni test). **b** mRNA expressions of six Cyp3a11 regulators in the small intestines of WT and *Bmal1⁻ᐟ⁻* mice at ZT2, 6, 10, 14, 18 and 22. Data are mean ± SD (*n* = 5). **P* < 0.05 for two group comparisons at individual time points (post hoc Bonferroni test). **c** Circadian Hnf4α and Dbp protein expressions in the livers of WT and *Bmal1⁻ᐟ⁻* mice. Data are mean ± SD (*n* = 5). **P* < 0.05 for two group comparisons at individual time points (post hoc Bonferroni test). **d** Circadian Hnf4α and Dbp protein expressions in the intestines of WT and *Bmal1⁻ᐟ⁻* mice. Data are mean ± SD (*n* = 5). **P* < 0.05 for two group comparisons at individual time points (post hoc Bonferroni test). **e** Effects of Bmal1 knockdown on Cyp3a11 regulators in Hepa1c1c7 cells. Data are mean ± SD (*n* = 5). **P* < 0.05 (*t* test). **f** Effects of Bmal1 knockdown on Cyp3a11 regulators in CT26 cells. Data are mean ± SD (*n* = 5). **P* < 0.05 (*t* test). Unshaded box represents the light period and the shaded box represents the dark (lights off) period

of *Cyp3a11* promoter (Fig. 3k), supporting the mediating roles of Hnf4α and Dbp in Bmal1 regulation of Cyp3a11. Taken together, Bmal1 regulated *Cyp3a11* expression through Dbp and Hnf4α.

**Bmal1 transcriptionally regulates Dbp and Hnf4α.** Bmal1 induced *Dbp* transcription in luciferase reporter assays (Fig. 4a). ChIP experiments showed significant Bmal1 recruitment to the E-box site of *Dbp* (Fig. 4b). These data confirmed Bmal1 as a transcriptional activator of *Dbp*[26]. Chromatin immunoprecipitation sequencing (ChIP-seq) revealed that Bmal1 bound to the distal region (from −6.1 to −6.0 kb) of the P1 promoter of *Hnf4α* (Fig. 4c). In luciferase reporter assays, Bmal1 efficiently induced *Hnf4α* P1 promoter (−6.1 kb) activity (Fig. 4d). The induction effect was lost when the distal region (from −6.1 kb to −6.0 kb) was truncated (Fig. 4d). Sequence analysis identified three potential E-box motifs (named E1, E2 and E3) within the 100-bp length of P1 promoter. EMSA assay further indicated direct binding of both E1 and E3 to Bmal1 protein in Bmal1-overexpressing Hepa1-6 cells (Fig. 4e and Supplementary Fig. 9).

However, no DNA−protein complex bands were generated when using nontransfected and Bmal1-lacking cells instead (Fig. 4e, f). ChIP assays confirmed that Bmal1 bound to the distal region of *Hnf4α* P1 promoter (Fig. 4g). Overall, our data support Bmal1 as a transcriptional activator of both Dbp and Hnf4α (Fig. 4h).

**Bmal1 modulates chronotoxicity of aconitine and triptolide via regulation of Cyp3a11.** Aconitine is a bioactive (and toxic) component of *Aconitum carmichaelii Debeaux*, a widely used herb medicine. Cyp3a11/CYP3A4 is the main enzyme responsible for aconitine metabolism and detoxification, generating four metabolites (Fig. 5a and Supplementary Fig. 10A/B)[27]. A high metabolic activity of Cyp3a11 toward aconitine was also confirmed using recombinant Cyp3a11 enzyme (Fig. 5b). Interestingly, the toxicity of aconitine (i.p. at 0.3 mg/kg or p.o. at 1.8 mg/kg) was dosing-time-dependent with a higher toxicity at daytime (ZT2/8) than at night (ZT14/20) (Fig. 5c, d). It was noted that a lower value of PR interval derived from ECG recordings is associated with high risks for preexcitation syndrome and supraventricular

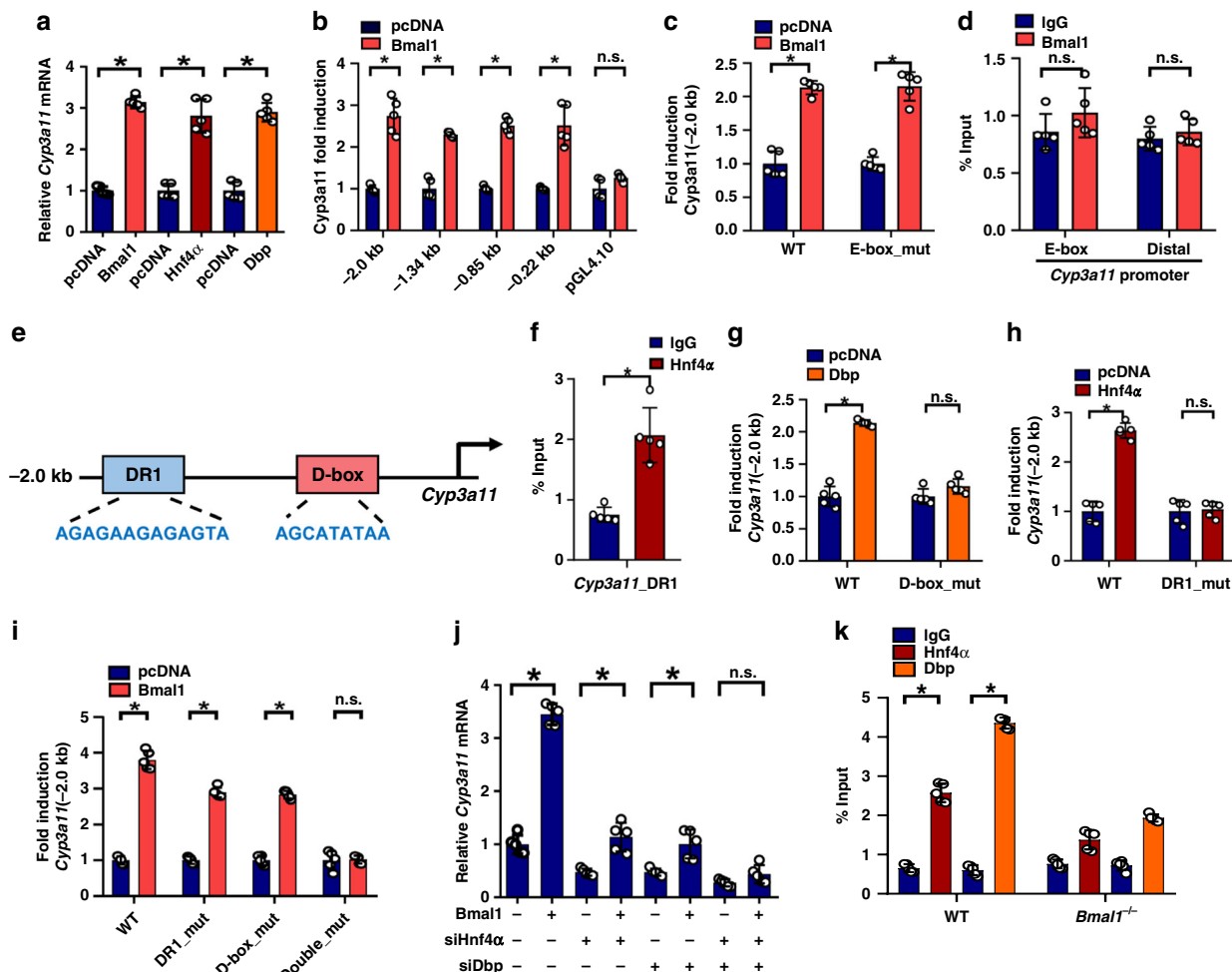

**Fig. 3** Dbp and Hnf4α mediate the regulation of Cyp3a11 by Bmal1. **a** Effects of Bmal1, Dbp and *Hnf4α* (P1 transcript) on *Cyp3a11* mRNA expression in Hepa-1c1c7 cells. Data are mean ± SD (*n* = 5). *P < 0.05 (*t* test). **b** Luciferase reporter assays with Hepa1-6 cells, showing the effects of Bmal1 on the activities of *Cyp3a11* reporters containing different versions of promoters. Data are mean ± SD (*n* = 5). *P < 0.05 (*t* test). **c** Luciferase reporter assays with Hepa1-6 cells, showing the effects of Bmal1 on the activities of WT or E-box-mutant *Cyp3a11* reporters (−2.0 kb). Data are mean ± SD (*n* = 5). *P < 0.05 (*t* test). **d** ChIP assays showing an interaction of Bmal1 with *Cyp3a11*-E-box region. Data are mean ± SD (*n* = 5). *P < 0.05 (*t* test). **e** The DNA sequences of DR1 and D-box present in the *Cyp3a11* promoter (−2.0 kb). **f** ChIP assays showing an interaction of Hnf4α with *Cyp3a11*-DR1. Data are mean ± SD (*n* = 5). *P < 0.05 (*t* test). **g** Effects of Dbp on the activities of *Cyp3a11* reporters in Hepa1-6 cells. Data are mean ± SD (*n* = 5). *P < 0.05 (*t* test). **h** Effects of Hnf4α on the activities of *Cyp3a11* reporters in Hepa1-6 cells. Data are mean ± SD (*n* = 5). *P < 0.05 (*t* test). **i** Effects of Bmal1 on the activities of −2.0 kb *Cyp3a11* reporters (WT, D-box mutant, DR1-mutant or Double-mutant) in Hepa1-6 cells. Data are mean ± SD (*n* = 5). *P < 0.05 (*t* test). **j** Effects of siHnf4α and/or siDbp on *Cyp3a11* expression in Hepa-1c1c7 cells overexpressing Bmal1. Data are mean ± SD (*n* = 5). *P < 0.05 (*t* test). **k** ChIP assays showing interactions of Hnf4α/Dbp with *Cyp3a11*-DR1/D-box in the livers of wild-type and *Bmal1*⁻/⁻ mice at ZT2. Data are mean ± SD (*n* = 5). *P < 0.05 (*t* test)

tachycardia[28]. This agreed well with the circadian protein expression and microsomal activity (Fig. 1d, e) of Cyp3a11 (higher levels at ZT14/20 and lower levels at ZT2/8). However, the time dependence of aconitine toxicity ceased to exist in Bmal1-deficient mice (Fig. 5e) consistent with blunted rhythms of Cyp3a11 expression and activity (Figs. 1d, e and 5f). Moreover, compared with wild-type, Bmal1-deficient mice showed higher levels of toxicity irrespective of dosing-time (Fig. 5e). This was probably associated with reduced metabolism of aconitine in Bmal1-deficient mice based on in vitro microsomal metabolism assays and pharmacokinetic analyses (Fig. 5f, g).

Triptolide is a bioactive (and toxic) component of *Tripterygium wilfordii Hook F*, an herbal medicine for treatment of leukemia and inflammation. Cyp3a11/CYP3A4 is the major enzyme contributing to triptolide metabolism and detoxification (Fig. 6a and Supplementary Fig. 10C&D)[29]. We confirmed that Cyp3a11 was highly active in catalyzing triptolide metabolism, generating

two hydroxylated metabolites (Fig. 6b). Similar to aconitine, triptolide toxicity showed dosing-time dependence with a more severe toxicity at daytime (ZT2/8) than at night (at ZT14/20). The diurnal variation in triptolide toxicity (Fig. 6c, d) was correlated with the circadian expression of Cyp3a11 (Fig. 1d, e). Bmal1 ablation resulted in a higher level of toxicity, and blunted its rhythmicity (Fig. 6c, d). This was accompanied by reduced levels of and dampened rhythm of triptolide metabolism based on microsomal metabolism assays and pharmacokinetic analysis (Fig. 6e, f). Taken together, Bmal1 modulates the chronotoxicity of aconitine and triptolide through regulation of rhythmic Cyp3a11.

**Discussion**

In this study, we clarified the impact of Bmal1 on generation of circadian Cyp3a11 expression, and elucidated the working

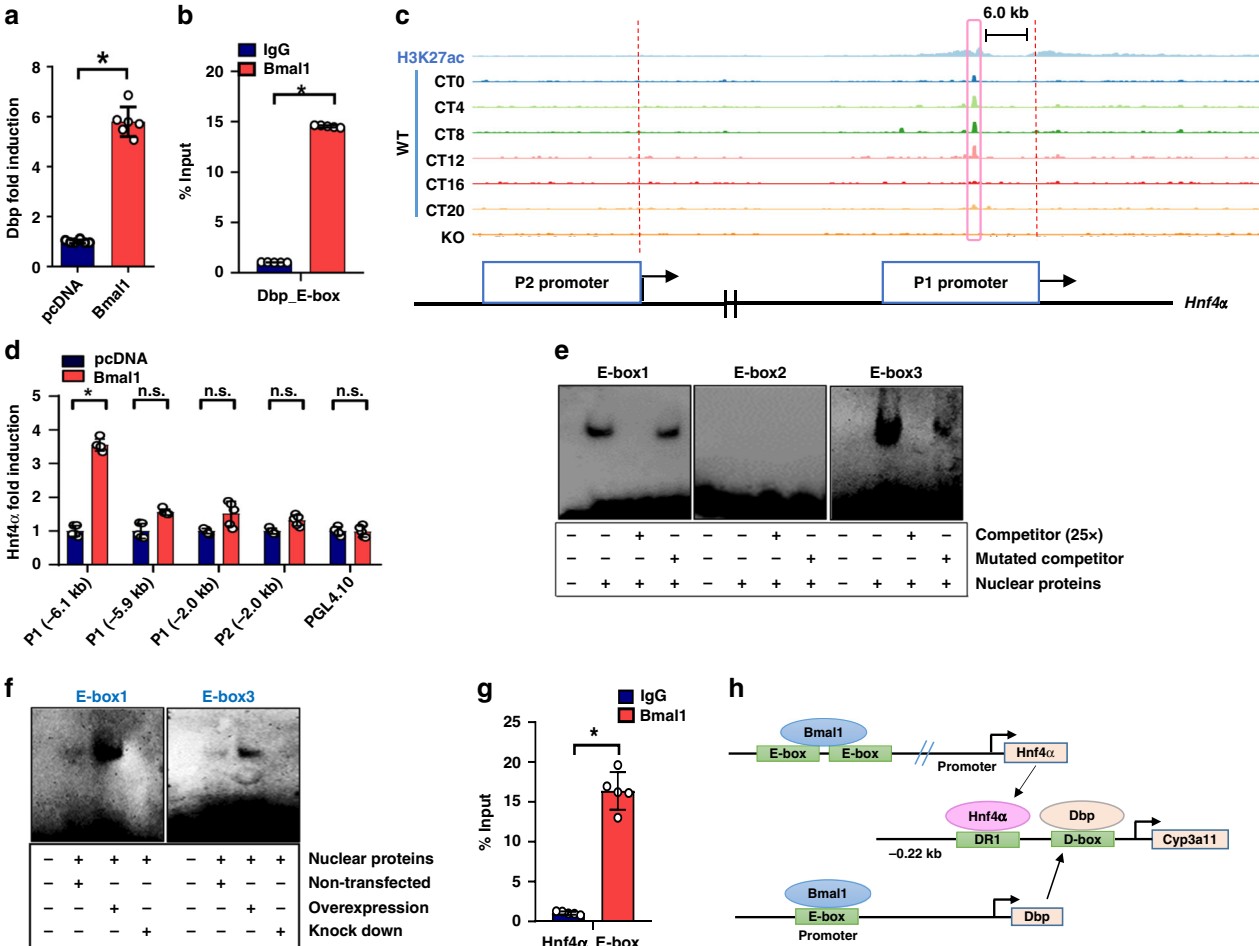

**Fig. 4** Bmal1 transcriptionally regulates Dbp and Hnf4α. **a** Bmal1 activates *Dbp* promoter in luciferase reporter assay in Hepa1-6 cells. Data are mean ± SD (*n* = 6 ). **P* < 0.05 (*t* test). **b** ChIP assays showing an interaction of Bmal1 with *Dbp* promoter in the livers of wild-type mice at ZT2. Immunoprecipitated chromatin was analyzed by quantitative real-time PCR with specific primers. **P* < 0.05 (*t* test). **c** Schematic diagram of ChIP-sequencing at six time points, showing that the signal of the *Hnf4α* P1 distal region  in WT mice (GSM982690), but not in *Bmal1*−/− mice (GSM982694). H3k27ac is a marker of transcriptional activation  (GSM982807). **d** Luciferase reporter assays in Hepa1-6 cells, showing the effects of Bmal1 on the activity of *Hnf4α* reporters containing different versions of promoters (i.e., distal region of P1 promoter (−6.1 kb), distal region of P1 promoter (−5.9 kb), P1 promoter (−2.0 kb) and P2 promoter (-2.0 kb)). Data are mean ± SD (*n* = 5). **P* < 0.05 (*t* test). n.s. not significant. **e** EMSA assays showing that Bmal1 binds to E-box 1 and E-box 3 in the distal region of *Hnf4α* P1 promoter (−6.1 kb~−6.0 kb). EMSA assay was performed with labeled E-box sites in the present or absent of nuclear extracts from Bmal1-overexpressed Hepa1-6 cells. **f** EMSA assays with nontransfected, Bmal1-overexpressed  and Bmal1 knocked-down Hepa1-6 cells, showing binding of Bmal1 to E-box 1 and E-box 3 in the distal region of Hnf4α P1 promoter (−6.1 kb~−6.0 kb). **g** ChIP assays showing an interaction of Bmal1 with *Hnf4α*-P1 E-boxes in the livers of wild-type mice at ZT2. Data are mean ± SD (*n* = 5). **P* < 0.05 (*t* test). **h** Schematic diagram showing the molecular mechanisms for Bmal1 regulation of Cyp3a11

mechanisms for Bmal1 regulation of Cyp3a11. Contrasting with a direct action on Sult1a1 (a phase II drug-metabolizing enzyme)[30], Bmal1 regulated Cyp3a11 using an indirect mechanism. This indirect mechanism involved two circadian genes *Dbp* and *Hnf4α* (both are positive regulators of Cyp3a11). Bmal1 transcriptionally activated Dbp and Hnf4α to induce Cyp3a11 expression (Fig. 4h). Dbp and Hnf4α exert an additive activation on Cyp3a11 transcription (Supplementary Fig. 11) and both are essential in mediating Bmal1 regulation of Cyp3a11 (Fig. 3). We also observed a tight association between Bmal1 function and drug (i.e., aconitine and triptolide) toxicity. Bmal1 ablation sensitized mice to drug toxicity (and blunted the circadian toxicity rhythmicity) due to reduced Cyp3a11-mediated metabolism and elevated drug exposure. Our study would facilitate the practice of chronotherapeutics (aiming to deliver drugs at a circadian time point with the maximal efficacy and least toxicity)[31].

To our knowledge, it is a previously unreported finding that Bmal1 transcriptionally activated the nuclear receptor Hnf4α (Figs. 2 and 4). The *Hnf4α* gene is known to have two promoters (named P1 and P2) that generate two different mRNA transcripts. *Hnf4α* P1 appears to be more important due to a superior activity in the recruitment of coactivators. *Hnf4α* P1 rather than P2 controls pancreatic β-cell function and has a tight association with metabolic disorders such as diabetes[32]. It was noteworthy that Bmal1 mainly acted on the P1 promoter  (Fig. 4c, d), generating circadian *Hnf4α* expression. Activation of P1 promoter to induce *Hnf4α* expression was also noted for other transcriptional factors such as *GR*, *HNF3* and *C/EBP*[33]. The possibility for Bmal1's action on P2 promoter was low because Bmal1 failed to induce the activity of P2 promoter (Fig. 4d).

To our knowledge, our study identified a previously unreported mechanism linking the core clock gene *Bmal1* with Cyp3a11, the foremost enzyme in drug metabolism. This regulatory axis

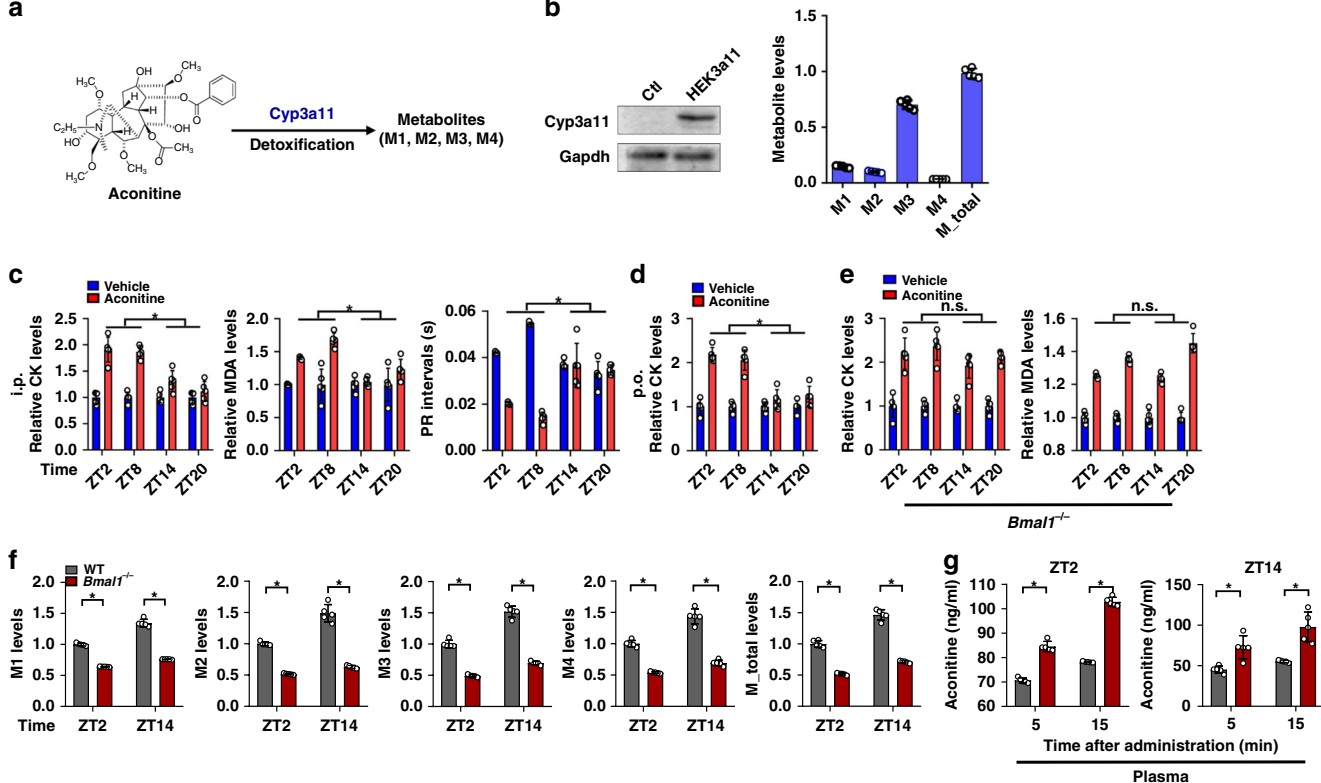

**Fig. 5** Bmal1 modulates circadian detoxification of aconitine via regulation of rhythmic Cyp3a11. **a** Main detoxification pathway for aconitine. The structures of M1, M2, M3 and M4 are provided in Supplemental Fig. 3A. **b** Metabolism of aconitine by recombinant Cyp3a11 prepared from Cyp3a11-overexpressing HEK293 (HEK3a11) cells. **c** Plasma CK and MDA levels as well as PR intervals in WT mice at 1 h after aconitine administration (0.3 mg/kg, i.p.) at ZT2, 8, 14 and 20. PR intervals represent the time from the onset of the P wave to the start of the QRS complex. Data are mean ± SD ($n = 5$). *$P < 0.05$ ($t$ test). **d** Plasma CK levels in WT mice at 1 h after aconitine administration (1.8 mg/kg, p.o.) at ZT2, 8, 14 and 20. Data are mean ± SD ($n = 5$). *$P < 0.05$ ($t$ test). **e** Plasma CK and MDA levels in $Bmal1^{-/-}$ mice at 1 h after aconitine administration (0.3 mg/kg, i.p.) at ZT2, 8, 14 and 20. Data are mean ± SD ($n = 5$). *$P < 0.05$ ($t$ test). **f** Generation of aconitine metabolites by liver microsomes prepared from WT and $Bmal1^{-/-}$ mice at ZT2 and ZT14. Data are mean ± SD ($n = 5$). *$P < 0.05$ ($t$ test). **g** Plasma aconitine concentrations at 5 and 15 min after intraperitoneal injection of aconitine to WT and $Bmal1^{-/-}$ mice at different circadian time points (ZT2 and ZT14). Data are mean ± SD ($n = 5$). *$P < 0.05$ ($t$ test)

Bmal1-Dbp/Hnf4α-Cyp3a11 underpinned the crosstalk between circadian clock and drug metabolism/chronotoxicity. It was noted that transcriptional regulation of Dbp by Bmal1, and of Cyp3a11 by Dbp/Hnf4α were separately reported in other studies[26,34,35]. Surprisingly, the DR1 element (−123/−110 bp) in *Cyp3a11* promoter for Hnf4α binding herein differed from that (−1581/−1524 bp) in the study of Inoue et al.[34]. Hnf4α was not responsive to −1581 bp DR1 element in our reporter assays (Supplementary Fig. 7). The exact reasons for this unexpected result were unknown. However, contrasting with the experimentation with human HepG2 cells in Inoue study[34], our experiments basing on mouse hepatoma cells were more appropriate for identification of the regulatory elements for mouse Cyp3a11.

We additionally determined the relative expressions of *Tef* and *Hlf* in wild-type vs. *Bmal1*$^{-/-}$ mice because they were suggested to be involved in regulation of xenobiotic detoxification together with Dbp in a previous study of Gachon et al.[20]. The roles of Tef and Hlf in Bmal1 regulation of Cyp3a11 were excluded as their expressions were unaffected in *Bmal1*$^{-/-}$ mice (Supplementary Fig. 6). This was also supported by the study of Oishi et al. in which Dbp rather than Tef and Hlf was downregulated in Clock-deficient mice[35]. In fact, in Gachon study there are no data supporting regulation of Cyp3a11 by Tef and Hlf although they may regulate the expressions of certain Cyp enzymes such as Cyp2b10[20]. Combined with these data, regulator screening

(Fig. 2a, b) and mechanistic investigations (Figs. 3 and 4) support Dbp and Hnf4α as the main clock-controlled regulators of Cyp3a11.

It was noteworthy that the circadian Cyp3a11 activity levels (probed by testosterone and midazolam) were not in a full agreement with circadian Cyp3a11 protein levels. This was probably because neither testosterone nor midazolam is a pure selective substrate of Cyp3a11. In addition to Cyp3a11, other Cyp enzymes such as Cyp1a2, Cyp3a13, Cyp7a1, Cyp7b1 and Cyp2b9 can metabolize testosterone albeit at lower activities[36]. Likewise, midazolam can be metabolized by the non-Cyp3a11 enzymes such as Cyp2c[37].

To our knowledge, it is a previously unreported discovery that *Hnf4α* was a clock-controlled gene and that Bmal1 participated in circadian regulation of Hnf4α. Bmal1 ablation blunted the expression rhythmicity of Hnf4α in mice (Fig. 2). Hnf4α is a key transcriptional regulator of many drug-metabolizing enzymes such as Cyps and Ugts[38]. Therefore, there is a high possibility that Hnf4α contributes to circadian rhythms of many other drug-metabolizing enzymes via its rhythmic expression and transcriptional regulation. This type of circadian regulation of a Cyp enzyme has been noted for the xenobiotic-sensing nuclear receptor Car[20]. Circadian expression of Cyp2b10 is accounted for by Car-mediated transactivation and its rhythmic expression[20].

Certain Cyp enzymes (e.g., Cyp2a4 and 2b10) display sex-specific expressions in the liver[39]. A recent study also reveals sex-

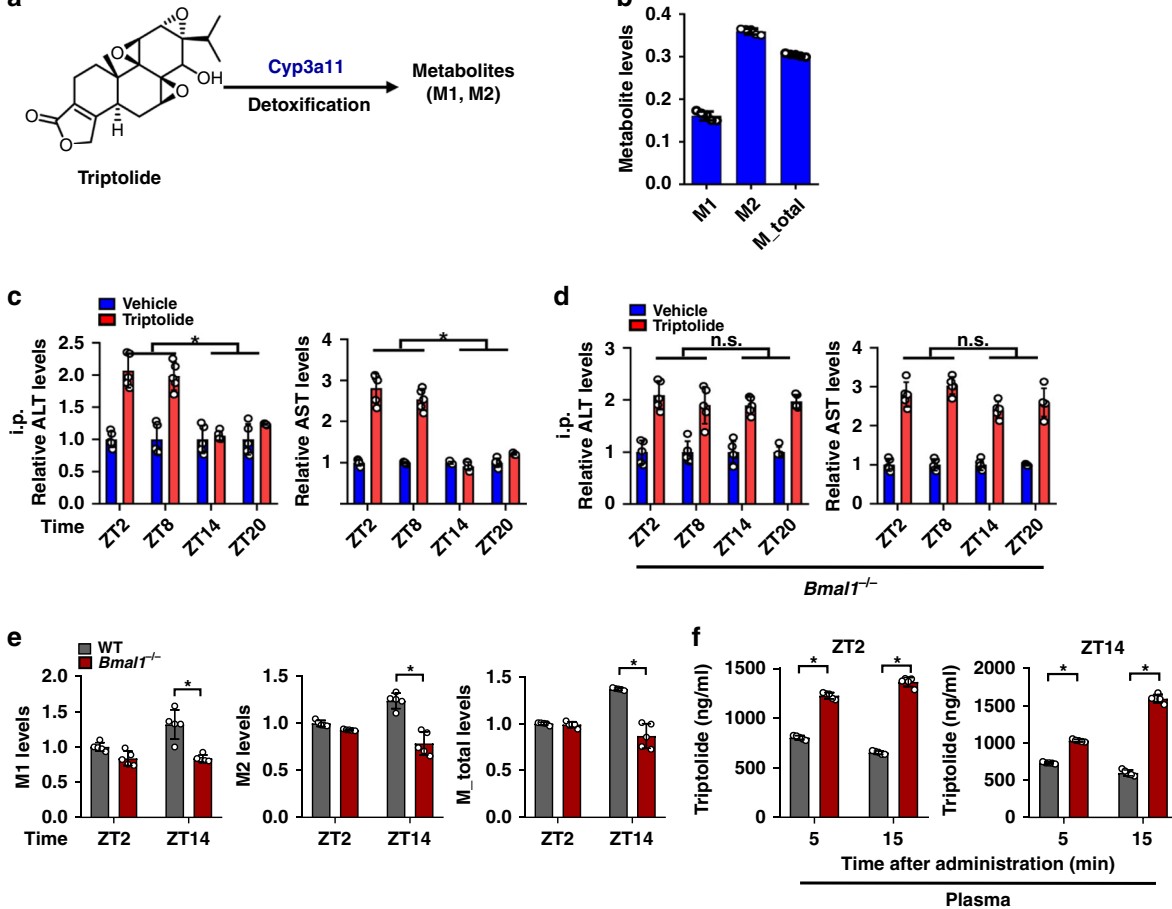

**Fig. 6** Bmal1 modulates circadian detoxification of triptolide via regulation of rhythmic Cyp3a11. **a** Main metabolic pathways for triptolide. **b** Metabolism of triptolide by recombinant Cyp3a11 prepared from Cyp3a11-overexpressing HEK293 (HEK3a11) cells. **c** Plasma ALT and AST levels in WT mice at 24 h after triptolide administration (0.6 mg/kg, i.p.) at ZT2, 8, 14 and 20. Data are mean ± SD ($n = 5$). *$P < 0.05$ ($t$ test). **d** Plasma ALT and AST levels in $Bmal1^{-/-}$ mice at 24 h after triptolide administration (0.6 mg/kg, i.p.) at ZT2, 8, 14 and 20. Data are mean ± SD ($n = 5$). *$P < 0.05$ ($t$ test). **e** Generation of triptolide metabolites by liver microsomes prepared from WT and $Bmal1^{-/-}$ mice at ZT2 and ZT14. Data are mean ± SD ($n = 5$). *$P < 0.05$ ($t$ test). **f** Plasma triptolide concentrations at 5 and 15 min after intraperitoneal injection of aconitine to WT and $Bmal1^{-/-}$ mice at different circadian time points (ZT2 and ZT14). Data are mean ± SD ($n = 5$). *$P < 0.05$ ($t$ test)

specific diurnal rhythms of gene expression and metabolism[40]. However, whether the diurnal Cyp3a11 rhythm is sex-dependent was not addressed here. In addition to gender, microbiome may be another influencing factor to inter-subject variations of circadian pattern in Cyp3a11[40]. We show Bmal1 controls circadian expressions of Cyp3a11 in the present study and of intestinal Mrp2 in our recent study[41]. BMAL1 has potential to regulate other key metabolic enzymes and transporters such as UGT1A1, CES2 and P-gp[42]. Together, these findings indicate a broad action of Bmal1 on xenobiotic metabolism and detoxification.

Preserved diurnal rhythms of certain genes and protein expressions were observed in $Bmal1^{-/-}$ mice (Fig. 2). This is not surprising because knockout of single core clock gene (e.g., Clock, Bmal1, Per2 and Rev-erbα) cannot completely abolish the rhythms of all circadian genes, but usually caused disruptions to circadian rhythms to a certain degree[43,44]. It is reasoned that circadian clock is a complicated system whose functions cannot be determined by only one of its components. We believe that a compensation mechanism from Bmal2 is trivial or none as the Bmal2 expression is unaffected in $Bmal1^{-/-}$ mice (Supplementary Fig. 12).

It was noteworthy that disrupted behavioral (feeding) rhythms may contribute to the loss of Cyp3a11 rhythms. This is because

food-derived chemicals may activate xenobiotic receptors such as PXR and CAR to induce Cyp3a11 expression. It is acknowledged that disrupted feeding rhythms might make a contribution to the loss of Cyp3a11 expression rhythms. We also observed downregulated P450 oxidoreductase (Por, a CYP redox partner) in $Bmal1^{-/-}$ mice consistent with the literature (Supplementary Fig. 13)[45]. Downregulation of Por may be additional potential mechanism for increased toxicity of aconitine and triptolide. However, whether this is true or not remains to be clarified.

$CYP3A4$ is the orthologous gene of $Cyp3a11$ in humans[46]. We provided in vitro evidence that human BMAL1 positively regulates $CYP3A4$ expression (Fig. 1g) consistent with the action of mouse Bmal1 on Cyp3a11. It is also noted that human $HNF4α$ and $DBP$ are transcriptional activators of $CYP3A4$[47,48]. Thus, it is tempting to propose that the regulatory axis of Bmal1-Dbp/ Hnf4α-Cyp3a11 (BMAL1-DBP/HNF4α-CYP3A4 for humans) may be also applicable to humans despite of the lack of sufficient human-related studies.

In summary, the core clock gene $Bmal1$ controls the circadian rhythmicity of Cyp3a11 and drug chronotoxicity through regulation of Dbp/Hnf4α. Our findings contribute extensively to a deep understanding of clock-controlled drug metabolism, and will facilitate the practice of chronotherapeutics.

## Methods

**Materials**. Hepa1-6 and HepG2 cells were purchased from American Type Culture Collection (Manassas, VA). Hepa1c1c7, CT-26, and Caco-2 cells were purchased from the cell bank of Chinese Academy of Sciences (Shanghai, China). The biochemical assay kits for CK, MDA, ALT and AST were purchased from Jiancheng Bioengineering Institute (Nanjing, China). BCA assay kit, cytoplasmic/nuclear protein extraction kit, and EMSA kit were purchased from Beyotime (Shanghai, China). Midazolam and NADPH were purchased from Sigma-Aldrich (St. Louis, MO). JetPrime transfection kit was purchased from POLYPLUS Transfection (Ill kirch, France). ChIP kit was purchased from Cell Signaling Technology (Beverly, MA). RNAiso Plus reagent and PrimeScript RT Master Mix were purchased from Takara (Shiga, Japan). ChamQ Universal SYBR qPCR Master Mix was purchased from Vazyme (Nanjing, China). Dual-Luciferase® Reporter Assay System was purchased from Promega (Madison, WI). Aconitine was purchased from MCE (MedChem Express, New Jersey). Quercetin, ketoconazole, vitamin K3, quinidine, 4-methylimidazole, testosterone and triptolide were purchased from Aladdin (Shanghai, China). pGL4.10 (see supplementary Fig. 1A for map) and pRL-TK vectors (see supplementary Fig. 1B for map) were purchased from Promega (Madison, WI). Cyp3a11-overexpressing HEK293 (named HEK3a11) cells, *Dbp* (2 kb)-Luc, pcDNA 3.1, pcDNA-Bmal1, pcDNA-Dbp, pcDNA-Hnf4α P1, and pcDNA-BMAL1 were obtained from Biowit Technologies (Shenzhen, China). The antibodies used in Western blotting were as follows: anti-Bmal1 (ab3350, Abcam, Cambridge, MA), anti-Cyp3a11 (ab3572, Abcam, Cambridge, MA), anti-Dbp (ab22824, Abcam, Cambridge, MA), anti-Hnf4α (TA321134, OriGene Technologies, Rockville, MD), and anti-Gapdh (ab9484, Abcam, Cambridge, MA). All the primary antibodies were diluted with 5% BSA at the ratio of 1:1000. The final concentration of antibodies was 1 μg/ml. The secondary antibody was purchased from Huaan Biotechnology and diluted with 5% skim milk at the ratio with 1:5000 (Hangzhou, China). For ChIP assay, anti-Bmal1 antibody was purchased from Abcam (Cambridge, MA) and normal rabbit IgG from Cell Signaling Technology (Beverly, MA). siRNA and negative control (siNC) were synthesized by Transheep Technologies (Shanghai, China).

**Plasmid construction**. Cyp3a11 (2 kb)-Luc, Cyp3a11 (1.34 kb)-Luc, Cyp3a11 (0.85 kb)-Luc, Cyp3a11 (0.22 kb)-Luc, Hnf4α-P1 (6.1 kb)-Luc, Hnf4α-P1 (5.9 kb)-Luc, Hnf4α-P1 (2.0 kb)-Luc, Hnf4α-P2 (2.0 kb)-Luc, Cyp3a11 (2 kb)-Luc containing mutated E-box, D-box or DR1 site, and Cyp3a11 (2 kb)-Luc containing double mutated D-box and DR1 sites were obtained from Transheep Technologies (Shanghai, China). The gene promoters were amplified by PCR reactions using specific primers (Supplementary Table 1). PCR amplification procedures consisted of an initial denaturation at 95 °C for 3 min, followed by 35 cycles of denaturation at 95 °C for 15 s, annealing at 60 °C for 30 s, and extension at 72 °C for 30 s, and a final extension at 72 °C for 5 min. Cyp3a11 promoter sequences were cloned into *Xho*I and *Bgl*II sites of pGL4.1 vector, and Hnf4α promoters into the *Xho*I and *Hind*III sites of pGL4.1 vector. All plasmids were verified by DNA sequencing. After transformed into *E. coli* JM109, plasmids were isolated using EasyPure HiPure Plasmid MiniPrep kits (TransGen Biotech, Beijing, China) according to the manufacturer's instructions.

**Animal studies**. All wild-type (C57BL/6J) and $Bmal1^{-/-}$ mice (on a C57BL/6J background) were maintained on a 12 h L:12 h D cycle (light on 7:00 a.m. to 7:00 p.m.), with free access to food and water. $Bmal1^{-/-}$ mice were generated by deleting a portion of exon 5 (Supplementary Fig. 2). Genotype of $Bmal1^{-/-}$ mice was verified in our previous publication[49]. $Bmal1^{-/-}$ mice showed a disrupted rhythmicity in wheel-running activity. The genetically modified mice were reproduced through breeding of heterozygous animals at expected Mendelian frequencies with no gender bias, and phenotypically normal in terms of body weight and survival at an age of ≤12 weeks. Experiments were planned and performed in accordance with current "3Rs" and ARRIVE guidelines and approved by Jinan University Institutional Animal Care and Use committee. We also applied the principles of the ARRIVE guideline for data management and interpretation, and all efforts were made to minimize the number of animals used and their suffering. Mice were randomly allocated into equal experimental groups based on body weight. Sample size was determined according to preliminary experiments, and experimenters were blinded for analysis purposes. For the analyses of mRNA, proteins and microsomal CYP enzyme activity, wild-type or $Bmal1^{-/-}$ mice (8–12 weeks of age, male, $n = 5$ per group) were sacrificed every 4 h (at ZT2, ZT6, ZT10, ZT14, ZT18 and ZT22, whereby ZT0 is defined as sunrise), and livers and small intestines were harvested.

Toxicological studies for aconitine and triptolide were performed with wild-type and $Bmal1^{-/-}$ mice as previously described[50]. The doses for aconitine and triptolide were 0.3 and 0.6 mg/kg, respectively. Plasma MDA (malonaldehyde), CK (creatine kinase), ALT (alanine aminotransferase), and AST (aspartate aminotransferase) levels were measured using the biochemical assay kits (Jiancheng Bioengineering Institute, Nanjing, China). Electrocardiogram (ECG) recordings of C57BL/6 mice treated with aconitine (i.p. at 0.3 mg/kg) at ZT2, ZT8, ZT14 or ZT20 were monitored using Dual Bio Amp/Stimulator (AD instruments, Sydney, Australia).

For pharmacokinetic studies, the drug (acontine (0.3 mg/kg) and triotplide (0.6 mg/kg)) was administered to wild-type or $Bmal1^{-/-}$ mice (8–12 weeks of age, male) by intraperitoneal injection at ZT2 or ZT14. At each time point, five mice were rendered unconscious with isoflurane for blood sampling. After centrifugation, the plasma samples were collected and subjected to LC/MS/MS analysis.

**Cell culture, treatment and transfection**. Hepa1-6, Hepa-1c1c7, CT-26, HepG2 and/or Caco-2 cells were used in the gene overexpression and knockdown experiments. Luciferase reporter, electrophoretic mobility shift and chromatin immunoprecipitation assays were based on Hepa1-6 cells. Hepa-1c1c7, CT26 and Caco-2 cells were cultured in DMEM medium supplemented with 10%FBS, Hepa1-6 cells in RPMI 1640 supplemented with 10%FBS, and HepG2 cells in the α-MEM medium supplemented with 10%FBS. Once reaching a confluence of 60–70%, the cells were transfected with indicated plasmids using a JetPrime transfection kit according to the manufacturer's protocol. Six hours later, the culture medium was replaced with fresh medium. On the next day, the cells were collected for further experiments.

**Mouse genotyping**. DNA was extracted from mouse tail (1–2 mm). PCR was performed with 400 ng DNA template. The products were subjected to the 2% agarose gel electrophoresis, and the bands were imaged using the Omega LumTM G imaging system (Aplegen). The primers are listed in Supplementary Table 2. Bands of 677 bp and 573 bp indicate wild-type and homozygous, respectively.

**qPCR**. qPCR assays were performed as previously described[13]. In brief, total RNA was extracted from mouse tissues and cells using RNAiso Plus reagent according to the manufacturer's instructions. 500 ng RNA was used as a template for cDNA synthesis using the PrimeScript RT Master Mix. The sequences of all primers applied to qPCR are listed in Supplementary Table 2. PCR amplification reactions consisted of an initial denaturation at 95 °C for 5 min, followed by 40 cycles of denaturation at 95 °C for 15 s, annealing at 60 °C for 30 s, and extension at 72 °C for 30 s. Mouse 18s RNA or human GAPDH was used as an internal control. Gene expression was calculated using the $2^{-\Delta\Delta CT}$ method. The primers (listed in Supplementary Table 2) were obtained from Ruiboxingke Biological Technology (Beijing, China). Primer specificity was checked by BLAST analysis (http://www.ncbi.nlm.nih.gov/BLAST/) and by performing a melting curve analysis. All qPCR assays were performed according to the MIQE (Minimum Information for Publication of qPCR experiments) guideline[51].

**Western blotting**. Mouse tissues or cells were lysed in RIPA buffer containing 1 mM PMSF. Protein concentrations were detected using a BCA assay kit. The samples were subjected to sodium dodecyl sulfate-polyacrylamide gel electrophoresis (10% acrylamide gels), and transferred onto PVDF membranes. After blocking with 5% skimmed milk for 1 h, the membranes were sequentially probed with the primary antibodies and horseradish peroxidase-conjugated secondary antibody. Protein bands were detected by enhanced chemiluminescence and imaged by the Omega LumTM G imaging system (Aplegen). The densities of protein bands were analyzed by Fluorchem 5500 software. Uncropped scans of representative blots are provided in Supplementary Fig. 14.

**In vitro metabolism assay**. Mouse liver/intestine microsomes and HEK3a11 cell lysate were prepared as previously described[13,52]. In vitro metabolic activity was determined using the published incubation procedures[53]. After 2-h-incubation at 37 °C, the reactions were terminated by adding ice-cold acetonitrile. The enzymatic activity was determined by LC/MS/MS analysis of the generated metabolites from substrates. Incubation conditions are shown in Supplementary Table 3. Preliminary experiments were performed to ensure that the rates of metabolism were determined under linear conditions with respect to incubation time and protein concentration.

**LC/MS/MS analysis**. Drugs and their metabolites were quantified using an LC/MS/MS system as previously described[50].

**Luciferase reporter assays**. Luciferase reporter assays were performed as previously described[14]. In brief, the cells were transfected with 250 ng of (Hnf4α or Cyp3a11) luciferase reporter plasmid, 50 ng of pRL-TK vector, and/or 250 ng (Bmal1) overexpression plasmid. After 6 h, the medium was changed to fresh medium. On the next day, the medium was removed and the cells were collected. The cells were lysed with 45 μl passive lysis buffer (Promega) and the luciferase activities were detected using the Dual-Luciferase® Reporter Assay system. Relative luciferase activity was expressed as the ratio of firefly over renilla luciferase activity.

**Chromatin immunoprecipitation assay (ChIP)**. ChIP assays were performed using the Enzymatic Chromatin IP kit as previously described[13]. In brief, mouse livers were fixed in 1% formaldehyde for 20 min at room temperature. The reactions were terminated by the addition of glycine. After 5 min centrifugation, the livers were collected and homogenized by using a Dounce homogenizer. Cells were incubated with ChIP buffer containing micrococcal nuclease. After addition of 0.5 M EDTA, the precipitate was suspended in ChIP buffer and subjected to sonication. An aliquot of sheared chromatin was used as Input sample, whereas the

remaining was immunoprecipitated with anti-Bmal1, anti-Hnf4α or anti-Dbp overnight at 4 °C. On the next day, the protein-DNA complex was incubated with protein G-coupled Dynabeads (30 μl) at 4 °C for 2 h. The immune complex was separated from beads and eluted in ChIP buffer. The immunoprecipitated chromatin was decross-linked at 65 °C for 4 h and protein was digest with proteinase K. The purified DNA was amplified by real-time PCR with specific primers (Supplementary Table 4).

**Electrophoretic mobility shift assay (EMSA)**. Nuclear proteins from the cells were first extracted using a cytoplasmic/nuclear protein extraction kit. EMSA assays were performed with a chemiluminescent EMSA kit as previously described[14]. In brief, 6-μg nuclear proteins were mixed with EMSA binding buffer in the presence or absence of 12.5 nmol unlabeled probes (or unlabeled mutated probes). The mixture was incubated at room temperature for 10 min. Then, the biotin-labeled probes (0.5 nmol) were added into the mixture and incubated at room temperature for 20 min. The DNA−protein complexes were loaded onto 4% nondenaturing polyacrylamide gels. After 30-min electrophoresis, the products were transferred onto Hybond-N + membranes for 45 min. The membranes were visualized by using enhanced chemiluminescence and Omega Lum G imaging system (Aplegen, Pleasanton, CA). The primers are listed in Supplementary Table 5.

**Statistical analysis and reproducibility**. Data are presented as mean ± SD (standard deviation). Statistical analyses on time-dependent data were performed using a two-way analysis of variance (ANOVA), followed by a post-hoc Bonferroni test (a post-hoc test was run only when a significant omnibus F-test was obtained) using GraphPad Prism software version 7 (GraphPad Software Inc., San Diego, CA). The homogeneity of variance assumption for ANOVA was verified by Levene's test. Statistical differences between two groups were analyzed by Student's t test. Circadian parameters were obtained through cosine curve fitting using the cosinor periodogram program (Boise State University, Boise, ID). The level of significance was set at $P < 0.05$ (*).

**Reporting summary**. Further information on research design is available in the Nature Research Reporting Summary linked to this article.

## Data availability
The data that support the findings of this study are available from the authors on reasonable request. The source data underlying plots are presented in Supplementary Data 1. Full blots are shown in Supplementary Information.

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

## Acknowledgements

This work was supported by the National Natural Science Foundation of China (Grant 81573488 and 81503210).

## Author contributions

B.W. and Y.L. designed the study; Y.L., S.W., Z.Z., L.G. and F.Y. performed experiments; Y.L., S.W. and Z.Z. collected and analyzed data; B.W., Y.L. and S.W. wrote the manuscript.

## Competing interests

The authors declare no competing interests.
