## [Peer Review File · Communications Biology]

Reviewers' comments:

Reviewer #1 (Remarks to the Author):

The paper claims that clock gene *Bmal1* controls circadian cytochrome P450 3a11, thus accounting for a major general mechanism of chronopharmacology. The study has been conducted in male mice and in cancer cell lines, using a variety of gene silencing techniques, bioluminescent reporters, and molecular biology determinations of mRNA and protein expression, and biochemical determinations of enzymatic activities.

The manuscript is clear, well presented, and the data that are presented in the Figures are generally supporting the scientific statements.

At first however the novelty of the finding was not obvious, as Gachon et al. (2006; cited ref 46) had first reported the indirect control of Phase1-3 metabolism by the molecular clock through the control of DBP, TEF and HLF in mice, and others had shown the role of clock genes *Clock*, *Bmal1* and *Per*'s on the chronotoxicity of drugs whose detoxification involved *Cyp3a11*. However through reading this manuscript, there is an elegant demonstration that *Bmal1* controls DBP and HNF4alpha, which in turn both control *Cyp3a11* transcription, resulting in circadian changes in both protein levels and enzymatic activity both in liver and in small intestine. The basic scientific question remaining, and which was addressed here should thus be clearly formulated beyond Gachon's and others' works.

The data convincingly show the selectivity of DBP and HNF4alpha rhythms regarding *Cyp3a11* rhythmic mRNA, protein and enzymatic activity in liver and small intestine of male C57/Bl6 mice, and their implications for the metabolism of *Cyp3a11* substrates midazolam and testosterone. Furthermore, *Bmal1* silencing or overexpression respectively reduced or increased *Cyp3a11* mRNA expression in mouse and human cancer cell lines, thus pinpointing the critical role of *Bmal1* for *Cyp3a11* rhythmic control.

However, the the mRNA and protein patterns of the transcription factors *Dbp*, *Hnf4alpha*, and *Car*, *Vdr*, *Fxr*, and those of other clock-controlled genes in the liver and small intestine of *Bmal1*^{-/-} mice need further statistical validation, and further explanations (Figure 2). It is surprising to see that *Bmal1*^{-/-} mice displayed apparently persistent 24-hour changes for most gene and protein expressions, with seemingly dampened amplitudes and occasional phase shifts. Cosinor analysis would be helpful to document whether the rhythms were suppressed or dampened and phase-shifted in the *Bmal1*^{-/-} mice. More should be said in the Methods or the introduction regarding the phenotype of the *Bmal1*^{-/-} mice. Is there any compensatory genetic mechanism – *Bmal2*, for instance? Or other?

Finally, the implications for the chronopharmacology of two *Cyp3a11* substrates are highlighted in wt or *Bmal1*^{-/-} mice using aconitine and triptolide as compounds of interest. The reason why only two timepoints were selected for this test needs to be explained. As a consequence, the lack of any difference between ZT2 and ZT14 dosing in *Bmal1*^{-/-} mice cannot be interpreted as a suppression of the chronopharmacology but its alteration, as it is

unknown whether Bmal1^{-/-} results in a rhythm phase shift or suppression.

Finally, the Discussion could address possible sex-related differences (Weger et al Cell Metabolism 2019), as well as Bmal1 control – direct or indirect) of other key metabolic pathways such as carboxylesterases, UGT's and ABC transporters, that impacted on irinotecan cellular pharmacokinetics in synchronized cell cultures (Dulong et al. Mol Cancer Ther 2015). They could also discuss mechanisms that could result in inter subject variations of circadian patterns in Cyp3a11 in humans,

Reviewer #2 (Remarks to the Author):

This manuscript describes the effects of loss of Bmal1 on Cyp3a11 gene regulation and subsequent effects on drug metabolism. The authors show that, in the absence of Bmal1, Cyp3a11 gene expression is repressed, with corresponding decreases in protein and drug-metabolizing activities. The authors propose that repression of Cyp3a11 expression occurs indirectly, through loss of Dbp and Hnf4a expression. They propose that Bmal1 transcriptionally activates Dbp and Hnf4a via E-box binding; these proteins then bind to Dr-1 and D-box sites in the Cyp3a11 promoter. Physiological significance of circadian expression of Cyp3a11 is demonstrated by observation of decreased metabolism and lack of circadian variation in toxicity in Bmal1 ^{-/-} animals.

There are a number of issues that must be addressed before this manuscript can be considered suitable for publication.

(1) Although Dbp has been shown to be downregulated in Bmal1 ^{-/-} mice, it has been reported that Hnf4a is not a transcriptional target of Bmal1 (Nuclear receptor HNF4A transrepresses CLOCK:BMAL1 and modulates tissue-specific circadian networks. Qu M, Duffy T, Hirota T, Kay SA. Proc Natl Acad Sci U S A. 2018 Dec 26;115(52):E12305-E12312. doi: 10.1073/pnas.1816411115. Epub 2018 Dec 10). The evidence for Bmal1 regulation of Hnf4a is shown in Figures 2 and 4. In Fig. 4D, binding to E-box 1 does not appear to be decreased by competitor and Fig. 4E is of poor quality and not informative. Although the text and legend refer to Hnf4a, Fig. 4F shows that Bmal1 binding to Dbp is much greater than to Hnf4a. The Western blots of Hnf4a protein indicating Hnf4a levels in Fig. 2 are also of poor quality. The lack of Por down-regulation in this knockout is in contrast to the downregulation observed in the liver-specific Bmal1 knockout.

(2) The description of methods needs improvement. The reference given for qPCR refers to another paper (Honma S, Kawamoto T, Takagi Y. et al. Dec1 and Dec2 are regulators of the mammalian molecular clock. Nature. 2002;419:841-844), which has no qPCR data in it. It appears that qPCR was done using SYBR Green but the method and the reagents are not mentioned. The authors should also refer to the MIQE guidelines for their assay and especially for validation of their probes. The details of the Chip assays are also sparse and it

is difficult to determine, for example, what the difference is between Hnf4a in Fig. 3D and the Hnf4a's in Fig. 4F.

Incubation times should be given for the P450 assays. Given the variation in Cyp3a11 levels, some demonstration of linearity over the concentrations used would be helpful.

Regarding ability of other researchers to replicate these results, I was unable to find English language information on reagents for two of the companies, Biowit and Bioraylabs. Details of the Bmal1 -/- mouse should be provided, such as location of the mutation and likelihood of truncated protein products. Did the authors look for circadian rhythm disruption in tissues other than liver and small intestine? Details of plasmids used should be provided, either by links to company websites or by inclusion of plasmid maps/sequences in supplementary methods.

(3) In general, more detail in the text and figure legends would be helpful. Many bands are not labelled, especially in the supplementary Western blots.

(4) There are numerous spelling and grammatical mistakes and the manuscript would benefit from language editing.

(5) The claim in the discussion that this is the first direct evidence for circadian clock controlled toxicity is should be removed.

Reviewer #3 (Remarks to the Author):

The authors utilize a genetics-based approach in order to investigate the role of the core clock component BMAL1 in driving the expression of Cyp3a11 and subsequent circadian rhythms of drug detoxification. They find that BMAL1 drives rhythmic expression of Cyp3a11 mRNA, protein, and activity in liver and intestine of WT mice. The authors identify regulatory elements in the Cyp3a11 promoter that are activated by DBP and HNF4a and conclude that the regulation of Cyp3a11 by BMAL1 is indirect. Finally, they demonstrate that ablation of Bmal1 in mice sensitizes mice to administered xenobiotics. Importantly, sensitivity of WT mice in these studies correlates with demonstrated circadian rhythms in Cyp3a11 protein expression and activity, which is abrogated in animals lacking Bmal1. This study adds significantly to the growing literature addressing the circadian regulation of xenobiotic metabolism. However, some of the conclusions are a bit overstated.

Points to Address:

1. The BMAL1-deficient mouse model requires some validation since it is not previously published. Authors should demonstrate loss of Bmal1 RNA and/or protein in the mice.
2. The use of global Bmal1-/- mice raises the possibility that indirect effects of abolishing

behavioral and physiological rhythms could contribute to the loss of Cyp3a11 expression rhythms. Since Cyp3a11 can be activated by the xenobiotic receptors PXR and CAR and they can be activated by many chemicals including phytoestrogens, mice must be fed a special semi-synthetic diet to avoid confounding effects of feeding time on measurements of Cyp3a11 RNA.

3. Conversely, it is very hard to understand how some of the molecular rhythms are preserved in global Bmal1^{-/-} animals. What do the authors propose is driving the circadian rhythms of expression of Tef, Hlf, Por, and Alas1 in the livers and intestines of Bmal1^{-/-} animals, which have no circadian rhythms in any tissue nor in feeding or activity? This is especially difficult to resolve with the fact that Por rhythmic expression is abolished in the liver of mice in which Bmal1 is deleted only in hepatocytes (Lamia et al PNAS 2008).

4. Some additional data presented conflict with prior literature and are difficult to understand. For example, the timing of rhythmic expression for many of the transcripts in this manuscript differ significantly from their expression patterns in other papers. See Koike et. al. (2012) supplementary data sets in table S7 where the peak of Dbp expression is ~CT8 rather than ~CT14 here.

5. Koike et al. identified a chromatin-binding site for endogenous BMAL1 at ~-1.5kb from the TSS in the Cyp3a11 locus in mouse livers. The experiment shown in Figure 3B does not exclude the possibility that this is an important site for direct activation of Cyp3a11 expression by BMAL1. The only expression construct that abolishes activation by BMAL1 is the one with no promoter at all so if any of the elements are present and conditions are saturating, it is not possible to identify specific active elements. To demonstrate that only the DR1 and D-box elements are required for activation by BMAL1, the experiment shown in Fig 4G needs to be repeated with the mutations introduced into the full length promoter (-2.0kb) element. None of the data exclude the possibility that BMAL1 acts both directly and indirectly to regulate Cyp3a11. A similar issue is also relevant for Supplementary Figure 6 regarding the HNF4a responsive elements.

6. Figures 4D/E are making the point that BMAL1 is being recruited to the Hnf4a promoter, however, Figure 4D only shows that some nuclear protein in the context of BMAL1 overexpression is binding the promoter elements, not necessarily BMAL1 itself. While 4E does demonstrate that lack of BMAL1 overexpression reduces binding of this protein to the promoter, this could also be explained by a protein induced by BMAL1 binding this region; overall, these experiments could be strengthened by performing this assay in cells lacking BMAL1 (both nontransfected and overexpressing BMAL1)

7. In Fig 4F, the divided y-axis obscures the first column of data – it is unclear how high is the control IgG signal for the Dbp-Ebox.

Minor Issues:

1. Several spelling/grammatical errors: Page 4 (CLCOK should be CLOCK); Page 5 ("robustness of circadian circuit" should be "circuits"); Page 15 ("PCR genotyping and expressing profiling" should be "expression"). The manuscript would benefit from additional editing for language and grammar.

2. Page 17 cites ChIP experiments showing Bmal1 recruitment to the E-Box of Dbp as Figure 4E, but this is not the correct figure.
3. Figure 4G is not cited in the text.
4. Some figures are cited out of order, including Supplementary Fig 5.
5. It has been demonstrated that CRY repressors affect expression of Cyp3a11 in mouse livers, which should be cited. (Kriebs et al PNAS 2017).

We wish to thank the reviewers for their careful and valuable reviews. Below are our point-by-point responses to the comments.

Reviewer #1 (Remarks to the Author):

The paper claims that clock gene Bmal1 controls circadian cytochrome P450 3a11, thus accounting for a major general mechanism of chronopharmacology. The study has been conducted in male mice and in cancer cell lines, using a variety of gene silencing techniques, bioluminescent reporters, and molecular biology determinations of mRNA and protein expression, and biochemical determinations of enzymatic activities.

The manuscript is clear, well presented, and the data that are presented in the Figures are generally supporting the scientific statements.

At first however the novelty of the finding was not obvious, as Gachon et al. (2006; cited ref 46) had first reported the indirect control of Phase1-3 metabolism by the molecular clock through the control of DBP, TEF and HLF in mice, and others had shown the role of clock genes Clock, Bmal1 and Per's on the chronotoxicity of drugs whose detoxification involved Cyp3a11. However through reading this manuscript, there is an elegant demonstration that Bmal1 controls DBP and HNF4alpha, which in turn both control Cyp3a11 transcription, resulting in circadian changes in both protein levels and enzymatic activity both in liver and in small intestine. The basic scientific question remaining, and which was addressed here should thus be clearly formulated beyond Gachon's and others' works.

Response: The reviewer shows a little concern about the novelty of our study. He/she mentioned that Gachon et al first reported the indirect control of phase1-3 metabolism by the molecular clock through the control of DBP, TEF and HLF in mice (Cell Metab. 2006;4(1):25-36). He/she also pointed out others had shown the role of clock genes Clock, Bmal1 and Per on the chronotoxicity of drugs whose detoxification involved Cyp3a11. It is true that Gachon et al in 2006 reported that the clock-output genes DBP, TEF, and HLF control Cyp2b10 expression thus modulate xenobiotic detoxification. The reviewer is also correct that clock genes Clock, Bmal1 and Per2 were previously reported to regulate the

chronotoxicity of drugs (cyclophosphamide and acetaminophen) (Proc Natl Acad Sci U S A. 2005; 102(9): 3407–3412 // Exp Toxicol Pathol. 2011;63(6):581-5). However, the major enzymes for detoxification or bio-activation are Cyp2b10, 2c29 and 3a13 (for cyclophosphamide) and are Cyp1a2 and 2e1 (for acetaminophen) with little or no contributions from Cyp3a11 (Proc Natl Acad Sci U S A. 2005; 102(9): 3407–3412 // Exp Toxicol Pathol. 2011;63(6):581-5). The findings from these previous studies DID suggest clock-controlled detoxification, although cyclophosphamide's chronotoxicity was independent of metabolism in the PNAS paper. However, whether and how the clock machinery (and its clock components) controls drug-metabolizing enzymes remained unknown. One of more important questions awaits investigations: whether and how Cyp3a11/CYP3A4 (the most important enzyme for drug metabolism/detoxification) is controlled by the clock system.

As commented by the reviewer that “through reading this manuscript, there is an elegant demonstration that Bmal1 controls DBP and HNF4alpha, which in turn both control Cyp3a11 transcription, resulting in circadian changes...”, we believe that current study is of high novelty because (1) a novel molecular link was uncovered between the core clock gene Bmal1 and Cyp3a11/CYP3A4 (the leading drug detoxification enzyme), (2) Cyp3a11 metabolism-mediated chronotoxicity was disclosed for two drugs aconitine and triptolide, (3) Hnf4a was newly identified as a circadian gene and Bmal1 transcriptionally regulates Hnf4a, accounting for its circadian expression. Clearly, all these were not resolved previously. Our findings contribute extensively to a deep understanding of clock-controlled drug metabolism, and will facilitate the practice of chronotherapeutics.

The reviewer raised another comment that scientific question to be addressed should be clearly formulated beyond Gachon's and others' works. In fact, it was stated in the INTRODUCTION section (the third paragraph) that” However, whether and how the clock machinery (and its clock components such as Bmal1 and Clock) controls drug-metabolizing enzymes remained unknown. One of more important questions awaits investigations: whether and how Cyp3a11/CYP3A4 (the most important enzyme for drug

metabolism/detoxification) is controlled by the clock system". It should be clear that current study deals with molecular links between the clock system (Bmal1) and drug metabolism/toxicity that differs from previous studies. Our newly defined regulatory axis of Bmal1-Dbp/Hnf4 α -Cyp3a11 (underpinning the crosstalk between circadian clock and drug metabolism/chronotoxicity) well explains the diurnal variations in drug toxicity, therefore significantly advances the sciences of detox biology and chronopharmacology .

The data convincingly show the selectivity of DBP and HNF4 α rhythms regarding Cyp3a11 rhythmic mRNA, protein and enzymatic activity in liver and small intestine of male C57/BL6 mice, and their implications for the metabolism of Cyp3a11 substrates midazolam and testosterone. Furthermore, Bmal1 silencing or overexpression respectively reduced or increased Cyp3a11 mRNA expression in mouse and human cancer cell lines, thus pinpointing the critical role of Bmal1 for Cyp3a11 rhythmic control. However, the mRNA and protein patterns of the transcription factors Dbp, Hnf4 α , and Car, Vdr, Fxr, and those of other clock-controlled genes in the liver and small intestine of Bmal1^{-/-} mice need further statistical validation, and further explanations (Figure 2). It is surprising to see that Bmal1^{-/-} mice displayed apparently persistent 24-hour changes for most gene and protein expressions, with seemingly dampened amplitudes and occasional phase shifts. Cosinor analysis would be helpful to document whether the rhythms were suppressed or dampened and phase-shifted in the Bmal1^{-/-} mice. More should be said in the Methods or the introduction regarding the phenotype of the Bmal1^{-/-} mice. Is there any compensatory genetic mechanism – Bmal2, for instance? Or other?

Response: We agree with the reviewer that cosinor analysis should be performed on the circadian data in Figure 2 to ascertain the rhythmic changes caused by Bmal1 deletion. Accordingly, we have performed cosinor analyses on mRNA and protein patterns of the transcription factors using the cosinor periodogram software (Boise State University, Boise, ID, USA). The obtained parameters have been added to Supplementary Table 6/7. Based on the parameters, the rhythms of Hnf4 α and Dbp in both liver and small intestine

were significantly dampened (without significant phase-shifts) in Bmal1^{-/-} mice.

The reviewer seems to question the persistent diurnal changes of certain gene and protein expressions in Bmal1^{-/-} mice. The reviewer may be not clear that knockout of single core clock gene (e.g., Clock, Bmal1, Per2 and Rev-erba) cannot completely abolish the rhythms of all circadian genes, but usually caused disruptions to circadian rhythms to certain degrees (J Pharm Sci. 2017;106(9):2491-2498 // Sci Transl Med. 2016;8(324):324ra16 // Mol Metab. 2013;2(3):292-305. // Nature. 2012;485(7396):123-7). This is most likely because circadian clock (biological clock) is a complicated system whose functions can not be determined by only one of its components. We believe a compensation mechanism from Bmal2 is trivial or none as the Bmal2 expression is unaffected in Bmal1^{-/-} mice (Supplementary Figure 10). To alleviate the reviewer's concern, the above point has been discussed in DISCUSSION section.

We agree with the reviewer that more should be said regarding the phenotype of Bmal1^{-/-} mice. Accordingly, we have added several sentences in the METHOD section to describe the phenotype of Bmal1^{-/-} mice.

Finally, the implications for the chronopharmacology of two Cyp3a11 substrates are highlighted in wt or Bmal1^{-/-} mice using aconitine and triptolide as compounds of interest. The reason why only two timepoints were selected for this test needs to be explained. As a consequence, the lack of any difference between ZT2 and ZT14 dosing in Bmal1^{-/-} mice cannot be interpreted as a suppression of the chronopharmacology but its alteration, as it is unknown whether Bmal1^{-/-} results in a rhythm phase shift or suppression.

Response: We accepted the reviewer's criticism that more circadian time points need to be tested for the chronotoxicity studies. Accordingly, we have performed new toxicity studies with aconitine and triptolide after drug dosing at two time points (ZT8 and ZT20). The new data have been added to Figure 5 & 6. ZT2/ZT8 and ZT14/ZT20 were selected

because they represent the light and dark phases, respectively.

The reviewer raised a comment that the lack of difference between ZT2 and ZT14 in *Bmal1*^{-/-} mice cannot be interpreted as a suppression of the chronopharmacology. We disagree with the reviewer on this matter. In fact, the lack of toxicity difference was due to the loss of rhythms of *Cyp3a11* expression and metabolism/detoxification (Figure 1 & 5 & 6). To be specific, the *Cyp3a11* detoxification at ZT14 was significantly suppressed and the toxicity at ZT14 was exacerbated (Figure 1 & 5 & 6). As a consequence, the rhythm of toxicity was lost in *Bmal1*^{-/-} mice (Figure 5 & 6). Therefore, it may be of little concern that the lack of difference between ZT2 and ZT14 was suppression of chronotoxicity.

Finally, the Discussion could address possible sex-related differences (Weger et al Cell Metabolism 2019), as well as *Bmal1* control – direct or indirect) of other key metabolic pathways such as carboxylesterases, UGT's and ABC transporters, that impacted on irinotecan cellular pharmacokinetics in synchronized cell cultures (Dulong et al. Mol Cancer Ther 2015). They could also discuss mechanisms that could result in inter subject variations of circadian patterns in *Cyp3a11* in humans,

Response: We have followed the reviewer's suggestion, and added a paragraph (the third last paragraph, see below) to discuss the following three points: (1) possible sex-related differences; (2) *Bmal1* control of other key metabolic pathways; and (3) mechanism resulting in inter-subject variations of circadian pattern of *Cyp3a11*.

“Certain Cyp enzymes (e.g., *Cyp2a4* and *2b10*) display sex-specific expressions in the liver. A recent study also reveals sex-specific diurnal rhythms of gene expression and metabolism. However, whether the diurnal *Cyp3a11* rhythm is sex-dependent was not addressed here. In addition to gender, microbiome may be another influencing factor to inter-subject variations of circadian pattern in *Cyp3a11*. We show *Bmal1* controls circadian expressions of *Cyp3a11* in the present study and of intestinal *Mrp2* in our recent

study. BMAL1 has potential to regulate other key metabolic enzymes and transporters such as UGT1A1, CES2 and P-gp. Together, these findings indicate a broad action of Bmal1 on xenobiotic metabolism and detoxification. ”

Reviewer #2 (Remarks to the Author):

This manuscript describes the effects of loss of Bmal1 on Cyp3a11 gene regulation and subsequent effects on drug metabolism. The authors show that, in the absence of Bmal1, Cyp3a11 gene expression is repressed, with corresponding decreases in protein and drug-metabolizing activities. The authors propose that repression of Cyp3a11 expression occurs indirectly, through loss of Dbp and Hnf4a expression. They propose that Bmal1 transcriptionally activates Dbp and Hnf4a via E-box binding; these proteins then bind to Dr-1 and D-box sites in the Cyp3a11 promoter. Physiological significance of circadian expression of Cyp3a11 is demonstrated by observation of decreased metabolism and lack of circadian variation in toxicity in Bmal1 $-/-$ animals.

There are a number of issues that must be addressed before this manuscript can be considered suitable for publication.

(1) Although Dbp has been shown to be downregulated in Bmal1 $-/-$ mice, it has been reported that Hnf4a is not a transcriptional target of Bmal1 (Nuclear receptor HNF4A transrepresses CLOCK:BMAL1 and modulates tissue-specific circadian networks. Qu M, Duffy T, Hirota T, Kay SA. Proc Natl Acad Sci U S A. 2018 Dec 26;115(52):E12305-E12312. doi: 10.1073/pnas.1816411115. Epub 2018 Dec 10).

Response: The reviewer raised a comment that Hnf4a is not a transcriptional target of Bmal1 based on the PNAS paper by Qu et al. We have carefully read the PNAS paper. In fact, the PNAS paper elegantly demonstrated trans-repression of CLOCK: BMAL1 by

Hnf4a via physical interactions. The paper provided a single set of luciferase reporter data (luciferase reporter is driven by a tandem consensus sequences of Hnf4a binding sites or four DR1 units, SI Appendix Fig S1 in the PNAS paper) that show no effects of BMAL1:CLOCK on the transcriptional activity of Hnf4a. Apparently, this experiment can not exclude the possibility of Hnf4a as a target of BMAL1:CLOCK. By contrast, we have provided multiple lines of evidence that Bmal1 transcriptionally regulates Hnf4a via binding to the E-boxes in P1 enhancer based on different sets of experiments including ChIP-seq (Figure 4C), luciferase reporter assays (Figure 4D), EMSA (Figure 4E&F) and ChIP assays (Figure 4G).

The evidence for Bmal1 regulation of Hnf4a is shown in Figures 2 and 4. In Fig. 4D, binding to E-box 1 does not appear to be decreased by competitor and Fig. 4E is of poor quality and not informative.

Response: The reviewer raised a comment that binding to E-box 1 does not appear to be decreased by competitor. In fact, E-box1 binding to Bmal1 was indeed decreased by competitor based on quantitative densitometry. The reviewer also commented that Fig. 4E is of poor quality. To alleviate the reviewer's concern, we have re-performed Figure 4D/E experiments and the figure panels have been updated. The new Figure 4E data show a clear decrease of E-box1 to Bmal1 by competitor. We believe Figure 4F quality has been significantly improved.

Although the text and legend refer to Hnf4a, Fig. 4F shows that Bmal1 binding to Dbp is much greater than to Hnf4a.

Response: We regret mistakes were made in description of Fig 4F in the text and legend. We have separated Figure 4F to two panels: one is ChIP data for Dbp (new Figure 4B), and the other is for Hnf4a (new Figure 4G). The descriptions have been corrected

accordingly in the revised manuscript.

The Western blots of Hnf4a protein indicating Hnf4a levels in Fig. 2 are also of poor quality.

Response: We regret WB images for Hnf4a in Fig 2 are poor quality. We have re-performed WB experiments to measure Hnf4a protein levels. We believe the quality of new WB images has been significantly improved (please see new Supplementary Fig 2).

The lack of Por down-regulation in this knockout is in contrast to the downregulation observed in the liver-specific Bmal1 knockout.

Response: We regret for this inconsistency. We have re-sampled the livers from Bmal1^{-/-} mice and performed new qPCR assays on Por. The relevant data have been updated and show down-regulation of Por in Bmal1^{-/-} mice (please see Supplementary Fig 9) that agrees with prior literature.

(2) The description of methods needs improvement. The reference given for qPCR refers to another paper (Honma S, Kawamoto T, Takagi Y. et al. Dec1 and Dec2 are regulators of the mammalian molecular clock. Nature. 2002;419:841-844), which has no qPCR data in it.

Response: Thanks. The reference has been corrected.

It appears that qPCR was done using SYBR Green but the method and the reagents are not mentioned. The authors should also refer to the MIQE guidelines for their assay and

especially for validation of their probes.

Response: Thanks. Details of qPCR have been added to the MATERIAL and METHOD section (please see the materials and method of “qPCR”).

The details of the Chip assays are also sparse and it is difficult to determine, for example, what the difference is between Hnf4a in Fig. 3D and the Hnf4a's in Fig. 4F.

Response: We regret the method for ChIP assay was not clear. Accordingly, ChIP method has been significantly revised.

Incubation times should be given for the P450 assays. Given the variation in Cyp3a11 levels, some demonstration of linearity over the concentrations used would be helpful.

Response: We regret the incubation times were not provided. We have added this information into the method part. We accept the reviewer's criticism that metabolism reaction linearity should be described. In fact, preliminary experiments were performed to ensure that the rates of metabolism were determined under linear conditions with respect to incubation time and protein concentration. This information has been added to the METHOD section in revised manuscript.

Regarding ability of other researchers to replicate these results, I was unable to find English language information on reagents for two of the companies, Biowit and Bioraylabs. Details of the Bmal1 ^{-/-} mouse should be provided, such as location of the mutation and likelihood of truncated protein products.

Response: The English information for the two companies Biowit and Bioraylab can be

found at the websites <http://www.biowit.com/> and <http://www.bioraylab.com/>, respectively. The location of deleted section has been provided in Supplementary Fig1. It should be noted that Bmal1 protein product was not detected in Bmal1^{-/-} mice (Fig 1B&C).

Did the authors look for circadian rhythm disruption in tissues other than liver and small intestine?

Response: Yes. Circadian rhythms are disrupted in the colon in our previous study (Nat Commun. 2018; 9: 4246.).

Details of plasmids used should be provided, either by links to company websites or by inclusion of plasmid maps/sequences in supplementary methods.

Response: The plasmid primers have been added to supplementary Table 3.

(3) In general, more detail in the text and figure legends would be helpful. Many bands are not labelled, especially in the supplementary Western blots.

Response: Revised as suggested.

(4) There are numerous spelling and grammatical mistakes and the manuscript would benefit from language editing.

Response: We have carefully proof-read the manuscript. We believe the language has been significantly improved.

(5) The claim in the discussion that this is the first direct evidence for circadian clock controlled toxicity is should be removed.

Response: Revised as suggested.

Reviewer #3 (Remarks to the Author):

The authors utilize a genetics-based approach in order to investigate the role of the core clock component BMAL1 in driving the expression of Cyp3a11 and subsequent circadian rhythms of drug detoxification. They find that BMAL1 drives rhythmic expression of Cyp3a11 mRNA, protein, and activity in liver and intestine of WT mice. The authors identify regulatory elements in the Cyp3a11 promoter that are activated by DBP and HNF4a and conclude that the regulation of Cyp3a11 by BMAL1 is indirect. Finally, they demonstrate that ablation of Bmal1 in mice sensitizes mice to administered xenobiotics. Importantly, sensitivity of WT mice in these studies correlates with demonstrated circadian rhythms in Cyp3a11 protein expression and activity, which is abrogated in animals lacking Bmal1. This study adds significantly to the growing literature addressing the circadian regulation of xenobiotic metabolism. However, some of the conclusions are a bit overstated.

Points to Address:

1. The BMAL1-deficient mouse model requires some validation since it is not previously published. Authors should demonstrate loss of Bmal1 RNA and/or protein in the mice.

Response: We regret our statements were not clear in original submission. In fact, the Bmal1^{-/-} mouse model has been described in our previous publication (Nat Commun. 2018; 9: 4246.). In this study, we have validated this genetic model by wheel running test,

genotyping, expression profiling (Nat Commun. 2018; 9: 4246.). In revised version, we have added a sentence to cite the NC paper and to clarify that Bmal1 ^{-/-} were already validated. Even though, we have confirmed the loss of Bmal1 mRNA/protein in the liver and small intestine in Bmal1^{-/-} mice (Figure 1A-C). This set of data were probably overlooked by the reviewer.

2. The use of global Bmal1^{-/-} mice raises the possibility that indirect effects of abolishing behavioral and physiological rhythms could contribute to the loss of Cyp3a11 expression rhythms. Since Cyp3a11 can be activated by the xenobiotic receptors PXR and CAR and they can be activated by many chemicals including phytoestrogens, mice must be fed a special semi-synthetic diet to avoid confounding effects of feeding time on measurements of Cyp3a11 RNA.

Response: The reviewer raised a good comment that disrupted behavioral (feeding) rhythms may contribute to loss of Cyp3a11 rhythms. This is because the food-derived chemical may activate xenobiotic receptors such as PXR and CAR to induce Cyp3a11 expression. Current study focused on circadian transcriptional regulation of Cyp3a11 by Bmal1. We did not intend to study the effects of feeding rhythms on Cyp3a11 expressions. We believe that investigating the impact on feeding patterns on Cyp3a11 expression should be an independent work.

To alleviate the reviewer's concern, we have added several sentences in DISCUSSION section (the ninth paragraph) to acknowledge a possible contribution of disrupted feeding rhythms to the loss of Cyp3a11 expression rhythms (see below).

"It was noteworthy that disrupted behavioral (feeding) rhythms may contribute to the loss of Cyp3a11 rhythms. This is because food-derived chemicals may activate xenobiotic receptors such as PXR and CAR to induce Cyp3a11 expression. It is acknowledged that disrupted feeding rhythms might make a contribution to the loss of Cyp3a11 expression

rhythms.”

3. Conversely, it is very hard to understand how some of the molecular rhythms are preserved in global *Bmal1*^{-/-} animals. What do the authors propose is driving the circadian rhythms of expression of *Tef*, *Hlf*, *Por*, and *Alas1* in the livers and intestines of *Bmal1*^{-/-} animals, which have no circadian rhythms in any tissue nor in feeding or activity? This is especially difficult to resolve with the fact that *Por* rhythmic expression is abolished in the liver of mice in which *Bmal1* is deleted only in hepatocytes (Lamia et al PNAS 2008).

Response: The reviewer raised a comment that it is hard to understand preserved rhythms of certain genes in *Bmal1*^{-/-} mice. The reviewer may be not clear that knockout of single core clock gene (e.g., *Clock*, *Bmal1*, *Per2* and *Rev-erba*) cannot completely abolish the rhythms of all circadian genes, but usually caused disruptions to circadian rhythms to certain degrees (J Pharm Sci. 2017;106(9):2491-2498 // Sci Transl Med. 2016;8(324):324ra16 // Mol Metab. 2013;2(3):292-305. // Nature. 2012;485(7396):123-7). This is most likely because circadian clock (biological clock) is a complicated system whose functions can not be determined by only one of its components.

We accept the reviewer’s criticism that the supplementary Fig 9 (*Por* expression) is in conflict with the PNAS paper by Lamia et al (2008). Accordingly, we have re-sampled the mouse livers at six circadian time points and re-performed the qPCR assays. The new data show decreased *Por* levels and blunted rhythms consistent with the Lamia paper. Further, we have removed the relevant inappropriate descriptions in the DISCUSSION section.

4. Some additional data presented conflict with prior literature and are difficult to understand. For example, the timing of rhythmic expression for many of the transcripts in this manuscript differ significantly from their expression patterns in other papers. See

Koike et al. (2012) supplementary data sets in table S7 where the peak of Dbp expression is ~CT8 rather than ~CT14 here.

Response: The reviewer raised a comment that some data are in conflict with prior literature. He/she further stated that the peak of Dbp expression is CT8 that differ from CT14 reported here. In fact, the peak times for Dbp expression in current study are ZT6-ZT10 (please see Fig 2) which agrees with the peak time reported by Koike et al (Supplementary Table S8 instead of S7, Science, 2012). It was noteworthy that the measured parameters such as peak times from in vivo studies are subject to moderate or even large variations. Peak times usually varies between different laboratories (one of causes refers to the sampling time). For example, circadian Hnf4a expression peaks from ZT10 to ZT14 (Cell Mol Gastroenterol Hepatol. 2015; 1(6): 664–677 // Nat Commun. 2018; 9: 4349). Circadian Dbp expression peaks from ZT6 and ZT8 (Nat Commun. 2018 12; 9(1):4246 //J Biol Chem. 2016 28; 291(44): 23318–23329). Circadian Cyp3a11 expression peaks from ZT3 to ZT10 (Chronobiol Int. 2013; 30(9): 1135–1143.//J Biol Chem. 2016; 291(10):4913-27).

5. Koike et al. identified a chromatin-binding site for endogenous BMAL1 at ~-1.5kb from the TSS in the Cyp3a11 locus in mouse livers. The experiment shown in Figure 3B does not exclude the possibility that this is an important site for direct activation of Cyp3a11 expression by BMAL1. The only expression construct that abolishes activation by BMAL1 is the one with no promoter at all so if any of the elements are present and conditions are saturating, it is not possible to identify specific active elements. To demonstrate that only the DR1 and D-box elements are required for activation by BMAL1, the experiment shown in Fig 4G needs to be repeated with the mutations introduced into the full length promoter (-2.0kb) element. None of the data exclude the possibility that BMAL1 acts both directly and indirectly to regulate Cyp3a11. A similar issue is also relevant for Supplementary Figure 6 regarding the HNF4a responsive elements.

Response: The reviewer asked for more evidence for indirect activation of Cyp3a11 by Bmal1. He/she also asked for tests of the -1.5 kb region for potential direct activation of Cyp3a11 by Bmal1 based on ChIP data from Koike paper (Science, 2012, 338(6105):349-54.). ChIP-seq data showed a rather weak signal (a tiny peak) at the -1.5 kb region at CT8 (see below), suggesting a potential non-canonical E-box (-1488~-1482 bp) for Bmal1 binding. It should be noted that this signal may be a false positive because it shows up only at the single time point ZT8 when six time points were analyzed, that was in a conflict with a circadian time expression of Bmal1 (with high protein expressions at ZT6-ZT12).

(This figure is drawn based on the data from Koike paper (Science, 2012, 338(6105):349-54.)

Anyway, we agree these requests are essential to pinpoint the exact type of mechanism for Bmal1 regulation of Cyp3a11. Accordingly, we have performed new mutagenesis experiments (for Bmal1 actions) using the -2.0 kb length luciferase reporter. The results showed that a mutation of D-box or DR1 alone attenuated but was unable to abrogate Bmal1 mediated activation of Cyp3a11 reporter. A dual mutation of D-box and DR1 eliminated the responsiveness of Cyp3a11 reporter to Bmal1. By contrast, mutation of the non-canonical E-box showed no effect on the luciferase activity. We also performed new ChIP assays on the non-canonical E-box. The results showed no recruitment of Bmal1 protein to this non-canonical E-box. All these new data support indirection rather than direct activation of Cyp3a11 by Bmal1, and support no action of Bmal1 on the -1.5 kb region. The new data have been added to Figure3C-3I, and described in the main text.

In a similar manner, we have performed new mutagenesis experiments (for Hnf4a actions) using the -2.0 kb length luciferase reporter. The results showed that a mutation of DR1 abrogated Hnf4a mediated activation of Cyp3a11 reporter. The new data have been added to Figure3H in the revised version.

6. Figures 4D/E are making the point that BMAL1 is being recruited to the Hnf4a promoter, however, Figure 4D only shows that some nuclear protein in the context of BMAL1 overexpression is binding the promoter elements, not necessarily BMAL1 itself. While 4E does demonstrate that lack of BMAL1 overexpression reduces binding of this protein to the promoter, this could also be explained by a protein induced by BMAL1 binding this region; overall, these experiments could be strengthened by performing this assay in cells lacking BMAL1 (both nontransfected and overexpressing BMAL1)

Response: We agree with the reviewer that EMSA assays should be also performed in cells lacking BMAL1. Accordingly, we have performed new EMSA assays using Bmal1-deficient cells. The new data support BMAL1 recruitment to the Hnf4a promoter, and have been added to Fig 4.

7. In Fig 4F, the divided y-axis obscures the first column of data – it is unclear how high is the control IgG signal for the Dbp-Ebox.

Response: We have separated Fig 4F into two panels (new Fig 4B & 4G), one shows ChIP data for binding of Bmal1 protein to the E-box of Dbp gene, and the other shows ChIP data for binding of Bmal1 protein to the E-box of Hnf4a gene. Through this operation, the clarity problem of the first column of data was resolved.

Minor Issues:

1. Several spelling/grammatical errors: Page 4 (CLCOK should be CLOCK); Page 5 (“robustness of circadian circuit” should be “circuits”); Page 15 (“PCR genotyping and expressing profiling” should be “expression”).

Response: Thanks. Fixed.

The manuscript would benefit from additional editing for language and grammar.

Response: We have carefully proof-read the manuscript. We believe the language has been significantly improved.

2. Page 17 cites CHIP experiments showing Bmal1 recruitment to the E-Box of Dbp as Figure 4E, but this is not the correct figure.

3. Figure 4G is not cited in the text.

4. Some figures are cited out of order, including Supplementary Fig 5.

Response: Thanks. These problems have been fixed.

5. It has been demonstrated that CRY repressors affect expression of Cyp3a11 in mouse livers, which should be cited. (Kriebs et al PNAS 2017).

Response: Thanks for the information. Kriebs paper reports that CRY1/2 serve as corepressors for many NRs, thereby likely contributing to diurnal modulation of drug metabolism. We have followed the reviewer’s suggestion and cited this paper (Ref 21) in the INTRODUCTION section.

Reviewers' comments:

Reviewer #1 (Remarks to the Author):

The manuscript has been improved considerably. There is a need for some improvements in the concepts formalisation, and in English as well. For instance:

Introduction

P4 Line 1: living beings rather than things

Line 6 : biological clocks is not equivalent to circadian clocks it is the periodicities that discriminates circadian from ultradian or infradian clocks. Is better to say "circadian rhythms reproduced and regulated by circadian clocks..." and also line 9 to be corrected accordingly. Lines 10-12: there are several feedback circuits, not a single one .

P5, lines 10-14, please indicate the species used for the studies reported.

Lines 15-16: "the molecular mechanisms involved remain largely elusive" This sentence should be downed, as a lot has been learnt and will need to be learnt after this paper as well....

P6 line 1 you mean "molecular clock-controlled metabolism and detoxification"?

second par is redundant to some extent with an earlier section in the Introduction. Can you shorten it?

Methods

P12, lines 9-10: unclear sentence

P15: Needs to mention cosine analyses as well here.

Results

P19, third to last line: please indicate that PR comes from EKG record

Discussion:

It would be useful for the reader to know whether Dbp and hNF4alpha exert additive activation on Cyp3a11 transcription, and subsequent translation and activity. Are both transcription factors the only clock-controlled regulators of Cyp3a11 (TEF and HLF were ruled out in your study)

Reviewer #2 (Remarks to the Author):

This manuscript investigates links between circadian rhythm and drug metabolism. The findings include direct regulation of Dbp and Hnf4a by Bmal1 and indirect circadian control of Cyp3a11 expression by Dbp and Hnf4a. Circadian variation in drug metabolism has been reported in many instances. Circadian regulation of Cyp3a11 and Dbp have been reported previously. Hnf4a has not been identified previously as a direct target of Bmal1; previous evidence has suggested that it is not directly regulated by Bmal1. The authors have addressed many, but not all, of the reviewers' concerns. Specific concerns are:

(1) Regarding the Bmal1 $-/-$ mice, the manuscript states, "The genetic mice were reproduced at expected Mendelian frequencies with no gender bias, and phenotypically normal in terms of body weight and survival." However, previous reports have shown that Bmal1 $-/-$ mice exhibit arthropathy, decreased survival after about 12 weeks of age, and female infertility.

Although the authors indicate that Por expression is down-regulated (Suppl. Fig. 10), the pattern is different from that reported by Lamia et al. and Johnson et al. [Johnson et al. (2014) PNAS 111:18757]. These differences or similarities should be discussed.

(2) In general, the comment on circadian variation in acetaminophen toxicity is oversimplified. If the authors wish to bring this up, they should mention that circadian variation in acetaminophen toxicity is dependent upon a number of factors, including glutathione availability and Por expression, in addition to Cyp2e1 levels.

(3) Down-regulation of Por is a potential mechanism for decreased toxicity of the compounds in this manuscript, especially in light of the observation that "It was noteworthy that the circadian Cyp3a11 activity levels (probed by testosterone and midazolam) were not in a full agreement with circadian Cyp3a11 protein levels." Johnson et al (2014) have shown that decreased hepatotoxicity of acetaminophen in Bmal1 ^{-/-} animals at ZT12 is due to decreased P450 activation resulting from decreased Por activity.

(3) Qu et al. clearly demonstrated that Bmal1 does not regulate Hnf4a in a luciferase reporter system and Koike et al demonstrate that, at the RNA level, Hnf4a does not cycle. This conflict should at least be discussed; for example, is Hnf4a regulation cell-type specific? Evidence that Hnf4a cycles in wildtype animals should be included in the manuscript. Methods for demonstrating circadian oscillation and periodicity should be included in the Methods section.

(4) I was able to access the Biowit website but the English version of the Bioraylab website continues to be inaccessible. The Biowit website does not provide maps of the pGL4.1 and pRL-TK plasmids. Methods for construction of the luciferase plasmids, such as PCR amplification and/or additional bases added during cloning, are not described. I will defer to Nature Communications editorial policy to determine if the description of materials used is sufficient.

(5) I believe this sentence is incorrect? "In luciferase reporter assays, Bmal1 efficiently induced Cyp3a11P1 promoter (-6.0 kb) activity (Figure 4D)."

(6) The methods for the P450 assays state that the concentration of NADPH stock solution was 100 mM. Note that this concentration is higher than the solubility limit of NADPH (https://www.sigmaaldrich.com/content/dam/sigma-aldrich/docs/Sigma/Product_Information_Sheet/2/n6505pis.pdf).

(6) Testosterone is misspelled in Fig. 1. Some figure references in the rebuttal letter are incorrect - I believe most of the manuscript is correct but this should be checked.

Reviewer #3 (Remarks to the Author):

While some of my concerns were addressed by the authors' rebuttal, others are enhanced by their apparent misunderstanding of prior literature and/or the state of the field.

1. The BMAL1-deficient mouse model requires some validation since it is not previously published. Authors should demonstrate loss of Bmal1 RNA and/or protein in the mice.

While the authors include measurement of BMAL1 protein they should include measurements of mRNA which is more quantitative and more sensitive. It remains possible that BMAL1 is expressed in some locations and could contribute to the very surprising remaining rhythmicity of several transcripts.

2. The use of global *Bmal1*^{-/-} mice raises the possibility that indirect effects of abolishing behavioral and physiological rhythms could contribute to the loss of *Cyp3a11* expression rhythms. Since *Cyp3a11* can be activated by the xenobiotic receptors PXR and CAR and they can be activated by many chemicals including phytoestrogens, mice must be fed a special semi-synthetic diet to avoid confounding effects of feeding time on measurements of *Cyp3a11* RNA.

The authors state that the impact of feeding on *Cyp3a11* should be reserved for another study but this is not a question it is a demonstrated fact that *Cyp3a11* is regulated by xenobiotics in the diet. If their mice truly have no rhythmic feeding patterns, this by itself could explain the loss of rhythmic *Cyp3a11* expression. If these authors wish to claim a different mechanism to explain the loss of rhythm, they must exclude this possibility.

3. Conversely, it is very hard to understand how some of the molecular rhythms are preserved in global *Bmal1*^{-/-} animals. What do the authors propose is driving the circadian rhythms of expression of *Tef*, *Hlf*, *Por*, and *Alas1* in the livers and intestines of *Bmal1*^{-/-} animals, which have no circadian rhythms in any tissue nor in feeding or activity? This is especially difficult to resolve with the fact that *Por* rhythmic expression is abolished in the liver of mice in which *Bmal1* is deleted only in hepatocytes (Lamia et al PNAS 2008).

The authors' response to this is wholly inadequate. It is simply not true that "The reviewer may be not clear that knockout of single core clock gene (e.g., *Clock*, *Bmal1*, *Per2* and *Rev-erba*) cannot completely abolish the rhythms of all circadian genes, but usually caused disruptions to circadian rhythms to certain degrees (J Pharm Sci. 2017;106(9):2491-2498 // Sci Transl Med. 2016;8(324):324ra16 // Mol Metab. 2013;2(3):292-305. // Nature. 2012;485(7396):123-7). This is most likely because circadian clock (biological clock) is a complicated system whose functions can not be determined by only one of its components." While disrupting most single core clock genes does not completely abolish rhythms, genetic deletion or suppression of *Bmal1* does in fact abolish the vast majority of transcriptional rhythms. Kornmann et al PLoS Biology (2007) is the best evidence for this. Specifically as mentioned in my initial review, deletion of *Bmal1* only in hepatocytes completely abolishes rhythmic *Por* expression. Although the authors state that they have re-performed this experiment and that their new data "show decreased *Por* levels and blunted rhythms consistent with the Lamia paper", that is not the case. They still see very robust rhythmic expression of *Por* which is absolutely not consistent with prior literature including the lamia 2008 paper and the Kornmann 2007 paper.

4. Some additional data presented conflict with prior literature and are difficult to understand. For example, the timing of rhythmic expression for many of the transcripts in this manuscript

differ significantly from their expression patterns in other papers. See Koike et. al. (2012) supplementary data sets in table S7 where the peak of Dbp expression is ~CT8 rather than ~CT14 here.

I apologize I may have misread the peak time of Dbp in this manuscript.

5. Koike et al. identified a chromatin-binding site for endogenous BMAL1 at ~-1.5kb from the TSS in the *Cyp3a11* locus in mouse livers. The experiment shown in Figure 3B does not exclude the possibility that this is an important site for direct activation of *Cyp3a11* expression by BMAL1. The only expression construct that abolishes activation by BMAL1 is the one with no promoter at all so if any of the elements are present and conditions are saturating, it is not possible to identify specific active elements. To demonstrate that only the DR1 and D-box elements are required for activation by BMAL1, the experiment shown in Fig 4G needs to be repeated with the mutations introduced into the full length promoter (-2.0kb) element. None of the data exclude the possibility that BMAL1 acts both directly and indirectly to regulate *Cyp3a11*. A similar issue is also relevant for Supplementary Figure 6 regarding the HNF4a responsive elements.

6. Figures 4D/E are making the point that BMAL1 is being recruited to the *Hnf4a* promoter, however, Figure 4D only shows that some nuclear protein in the context of BMAL1 overexpression is binding the promoter elements, not necessarily BMAL1 itself. While 4E does demonstrate that lack of BMAL1 overexpression reduces binding of this protein to the promoter, this could also be explained by a protein induced by BMAL1 binding this region; overall, these experiments could be strengthened by performing this assay in cells lacking BMAL1 (both nontransfected and overexpressing BMAL1)

Okay.

7. In Fig 4F, the divided y-axis obscures the first column of data – it is unclear how high is the control IgG signal for the Dbp-Ebox.

Okay

Minor Issues: ALL OK.

1. Several spelling/grammatical errors: Page 4 (CLCOK should be CLOCK); Page 5 ("robustness of circadian circuit" should be "circuits"); Page 15 ("PCR genotyping and expressing profiling" should be "expression"). The manuscript would benefit from additional editing for language and grammar.
2. Page 17 cites ChIP experiments showing *Bmal1* recruitment to the E-Box of Dbp as Figure 4E, but this is not the correct figure.
3. Figure 4G is not cited in the text.
4. Some figures are cited out of order, including Supplementary Fig 5.
5. It has been demonstrated that CRY repressors affect expression of *Cyp3a11* in mouse

livers, which should be cited. (Kriebs et al PNAS 2017).

We wish to thank the reviewers again for their careful and valuable comments. Below are our point-by-point responses to the comments.

Reviewer #1

The manuscript has been improved considerably. There is a need for some improvements in the concepts formalisation, and in English as well. For instance:

Introduction

P4 Line 1: living beings rather than things.

Response: Revised as suggested.

Line 6: biological clocks is not equivalent to circadian clocks it is the periodicities that discriminates circadian from ultradian or infradian clocks. Is better to say" circadian rhythms reproduced and regulated by circadian clocks..." and also line 9 to be corrected accordingly.

Response: Thanks. Revised as suggested.

Lines 10-12: there are several feedback circuits, not a single one.

Response: Fixed.

P5, lines 10-14, please indicate the species used for the studies reported.

Response: The species has been specified.

Lines 15-16: "the molecular mechanisms involved remain largely elusive" This sentence should be downed, as a lot has been learnt and will need to be learnt after this paper as well...

Response: We accept the reviewer's criticism. Accordingly, we have removed the sentence "the molecular mechanisms involved remain largely elusive".

P6 line 1 you mean" molecular clock-controlled metabolism and detoxification"?

Response: Revised as suggested.

Second part is redundant to some extent with an earlier section in the Introduction. Can you shorten it?

Response: We have removed the first sentence "As noted above, BMAL1 and CLOCK form the positive limb of the feedback loop in generation of circadian gene expression" which may be redundant with the first paragraph (introducing the feedback loops of circadian clock).

Methods

P12, lines 9-10: unclear sentence.

Response: The sentence has been re-written.

P15: Needs to mention cosine analyses as well here.

Response: Revised as suggested.

Results

P19, third to last line: please indicate that PR comes from ECG recording.

Response: Revised as suggested.

Discussion:

It would be useful for the reader to know whether Dbp and HNF4alpha exert additive activation on Cyp3a11 transcription, and subsequent translation and activity. Are both transcription factors the only clock-controlled regulators of Cyp3a11 (TEF and HLF were ruled out in your study).

Response: We have followed the reviewer's suggestion, and discussed the two points raised (whether Dbp and HNF4alpha exert additive activation on Cyp3a11 transcription; are both transcription factors the only clock-controlled regulators of Cyp3a11). Please see the first and the fourth paragraphs in the DISCUSSION section of revised manuscript.

Reviewer #2

This manuscript investigates links between circadian rhythm and drug metabolism. The findings include direct regulation of Dbp and Hnf4a by Bmal1 and indirect circadian control of Cyp3a11 expression by Dbp and Hnf4a. Circadian variation in drug metabolism has been reported in many instances. Circadian regulation of Cyp3a11 and Dbp have been reported previously. Hnf4a has not been identified previously as a direct target of Bmal1; previous evidence has suggested that it is not directly regulated by Bmal1. The authors have addressed many, but not all, of the reviewers' concerns. Specific concerns are:

(1) Regarding the Bmal1 $-/-$ mice, the manuscript states, "The genetic mice were reproduced at expected Mendelian frequencies with no gender bias, and phenotypically normal in terms of body weight and survival." However, previous reports have shown that Bmal1 $-/-$ mice exhibit arthropathy, decreased survival after about 12 weeks of age, and female infertility.

Response: The reviewer raised a comment that our statements are inconsistent with the previous reports. In fact, the genetic mice at the age of 8-12 weeks were used for experiments and sacrificed after experimentation. The reported phenotypes were restricted to mice at age of ≤ 12 weeks, which are indeed consistent with the previous reports (Proc Natl Acad Sci U S A.;107(44):19090-5.// J Biol Rhythms.;23(1):26-36.// Cell Metab. 2015;22(3):448-59.). Apparently, the phenotypes of Bmal1 $-/-$ mice may be age-dependent. We regret our previous statement regarding the fertility was not clear. The reviewer is right that the male and female homozygotes are infertile. To alleviate the reviewer's concern, we have corrected the phenotype description to "The genetic mice were reproduced through breeding of heterozygous animals at expected Mendelian frequencies with no gender bias, and phenotypically normal in terms of body weight and survival at an age of ≤ 12 weeks."

Although the authors indicate that Por expression is down-regulated (Suppl. Fig. 10), the pattern is different from that reported by Lamia et al. and Johnson et al. [Johnson et al. (2014) PNAS 111:18757]. These differences or similarities should be discussed.

Response: We have gone back to original data, and identified some mistakes in data analyses of Por expression. We regret for this careless. The data have been fixed (please new Supplementary Figure 13). Por rhythmic expression is indeed abolished in the liver of Bmal1^{-/-} mice consistent with prior literature.

(2) In general, the comment on circadian variation in acetaminophen toxicity is oversimplified. If the authors wish to bring this up, they should mention that circadian variation in acetaminophen toxicity is dependent upon a number of factors, including glutathione availability and Por expression, in addition to Cyp2e1 levels.

Response: We accept the reviewer's criticism. This sentence regarding acetaminophen toxicity has been removed.

(3) Down-regulation of Por is a potential mechanism for decreased toxicity of the compounds in this manuscript, especially in light of the observation that "It was noteworthy that the circadian Cyp3a11 activity levels (probed by testosterone and midazolam) were not in a full agreement with circadian Cyp3a11 protein levels." Johnson et al (2014) have shown that decreased hepatotoxicity of acetaminophen in Bmal1^{-/-} animals at ZT12 is due to decreased P450 activation resulting from decreased Por activity.

Response: We agree with the reviewer that Down-regulation of Por may be a potential mechanism for increased (note that the reviewer made a mistake here, increased rather

than decreased toxicity because Cyp metabolism is detoxification mechanism for test compounds) toxicity of the compounds in this manuscript. Accordingly, we have discussed this point in the DISCUSSTION section (please see the ninth paragraph) in revised manuscript.

(3) Qu et al. clearly demonstrated that Bmal1 does not regulate Hnf4a in a luciferase reporter system and Koike et al demonstrate that, at the RNA level, Hnf4a does not cycle. This conflict should at least be discussed; for example, is Hnf4a regulation cell-type specific? Evidence that Hnf4a cycles in wildtype animals should be included in the manuscript. Methods for demonstrating circadian oscillation and periodicity should be included in the Methods section.

Response: As indicated in the editor's email, the authors may not need to respond to the comments on Hnf4a rhythmicity due to the reviewer's misunderstanding.

(4) I was able to access the Biowit website but the English version of the Bioraylab website continues to be inaccessible. The Biowit website does not provide maps of the pGL4.1 and pRL-TK plasmids. Methods for construction of the luciferase plasmids, such as PCR amplification and/or additional bases added during cloning, are not described. I will defer to Nature Communications editorial policy to determine if the description of materials used is sufficient.

Response: We have verified that the Bioraylab website (<http://www.bioraylab.com/>) is accessible, but regret that the English version is not completely written in English.

We have provided the maps for pGL4.1 and pRL-TK in the Supplementary Figure 1 in the revised manuscript (please note that pGL4.1 and pRL-TK are commercialized), and added the methods for the luciferase plasmid construction in the METHOD section in the

revised manuscript.

(5) I believe this sentence is incorrect? “In luciferase reporter assays, Bmal1 efficiently induced Cyp3a11P1 promoter (-6.0 kb) activity (Figure 4D).”

Response: Thanks. This has been fixed.

(6) The methods for the P450 assays state that the concentration of NADPH stock solution was 100 mM. Note that this concentration is higher than the solubility limit of NADPH

(https://www.sigmaaldrich.com/content/dam/sigma-aldrich/docs/Sigma/Product_Information_Sheet/2/n6505pis.pdf).

Response: The product information states that “ β -NADPH is soluble in 0.01 M sodium hydroxide (50 mg/ml=60.0 mM), yielding a clear, light yellow solution.” However, it has no information on the solubility of NADPH. We are pretty sure that a stock of 100 mM NADPH can be prepared as previously reported (Sci Rep. 2016; 6:23135.).

(6) Testosterone is misspelled in Fig. 1. Some figure references in the rebuttal letter are incorrect - I believe most of the manuscript is correct but this should be checked.

Response: This problem has been fixed. Thanks.

Reviewer #3

While some of my concerns were addressed by the authors' rebuttal, others are enhanced by their apparent misunderstanding of prior literature and/or the state of the field.

1. While the authors include measurement of BMAL1 protein they should include measurements of mRNA which is more quantitative and more sensitive. It remains possible that BMAL1 is expressed in some locations and could contribute to the very surprising remaining rhythmicity of several transcripts.

Response: We have followed the reviewer's suggestion, and added the Bmal1 mRNA data of WT and Bmal1^{-/-} mice at six time points (please see new Figure 1C). The results showed that Bmal1 transcript was absent in Bmal1^{-/-} mice, supporting successful knockout of Bmal1.

2. The use of global Bmal1^{-/-} mice raises the possibility that indirect effects of abolishing behavioral and physiological rhythms could contribute to the loss of Cyp3a11 expression rhythms. Since Cyp3a11 can be activated by the xenobiotic receptors PXR and CAR and they can be activated by many chemicals including phytoestrogens, mice must be fed a special semi-synthetic diet to avoid confounding effects of feeding time on measurements of Cyp3a11 RNA.

The authors state that the impact of feeding on Cyp3a11 should be reserved for another study but this is not a question it is a demonstrated fact that Cyp3a11 is regulated by xenobiotics in the diet. If their mice truly have no rhythmic feeding patterns, this by itself could explain the loss of rhythmic Cyp3a11 expression. If these authors wish to claim a different mechanism to explain the loss of rhythm, they must exclude this possibility.

Response: The reviewer raised the comments that if their mice truly have no rhythmic feeding patterns, this by itself could explain the loss of rhythmic Cyp3a11 expression, and that if these authors wish to claim a different mechanism to explain the loss of rhythm, they must exclude this possibility. We understand the reviewer's concern. It is true that feeding schedule resets the liver clock and affects the rhythmic expression of a number of genes (note that NOT all genes) (Proc Natl Acad Sci U S A. 2009;106(50):21453-8.//Curr Biol. 2017;27(12):1768-1775.e3.// Science. 2001;291(5503):490-3.). We may disagree with the reviewer's reasoning that feeding pattern contributes to diurnal Cyp3a11 expression through xenobiotic receptor stimulation by dietary chemicals.

We would like to draw the reviewer's attention to the following points:

- (1) Both circadian clock and feeding pattern drive rhythmic transcription (Proc Natl Acad Sci U S A. 2009;106(50):21453-8.). It was shown that 617 rhythmic transcripts were driven by food only, 368 rhythmic transcripts were driven by circadian clock only, whereas the rest were driven by both (Proc Natl Acad Sci U S A. 2009;106(50):21453-8.).
- (2) Fasting-refeeding does not alter Cyp3a11 expression (see below), and feeding consolidation does not restore rhythmicity of Cyp3a11 in Cry DKO mice (data retrieved from <http://circadian.salk.edu>. pMMC β <0.05, Proc Natl Acad Sci U S A. 2009;106(50):21453-8.).

- (3) Arrhythmic feeding (6-meals-a-day feeding) hardly changes diurnal gene expression of P450-mediated drug metabolism (i.e., daily gene expression of most aspects of P450-mediated drug metabolism still follows the expression pattern of rhythmic fed animals), indicating a factor different from meal timing affects circadian gene expression (de Vries et al. PLoS one.2017.12(10):e0185520).
- (4) Semi-synthetic (elemental) diet reduces intestinal monooxygenase activity probably due to the lack of inducing effects of dietary chemicals on mucosal monooxygenase (Gastroenterology. 1984;86(6):1519-30. Thanks the reviewer for providing this reference). Assuming food (dietary chemicals) induces Cyp3a11 expression, arrhythmic feeding would accompany a loss of Cyp3a11 rhythm, but may not lead to marked down-regulation of Cyp3a11 (as observed in current study, Figure 1) because of no significant difference in total food intake (Cyp3a11 stimulants).

Based on the above analyses, one can see that the effects of feeding pattern on diurnal expression are gene specific. Cyp3a11 is a circadian clock-controlled gene (already validated in current study) whose rhythmicity may be determined mainly by circadian clock rather than feeding time. However, without more specific experimental supports, we cannot exclude a possibility for diurnal contribution from feeding pattern. This was why we had added several sentences (see below) in the DISCUSSION section to acknowledge a possible contribution of disrupted feeding rhythms to the loss of Cyp3a11 expression rhythms.

“It was noteworthy that disrupted behavioral (feeding) rhythms may contribute to the loss of Cyp3a11 rhythms. This is because food-derived chemicals may activate xenobiotic receptors such as PXR and CAR to induce Cyp3a11 expression. It is acknowledged that disrupted feeding rhythms might make a contribution to the loss of Cyp3a11 expression rhythms.”

3. The authors' response to this is wholly inadequate. It is simply not true that "The reviewer may be not clear that knockout of single core clock gene (e.g., Clock, Bmal1, Per2 and Rev-erba) cannot completely abolish the rhythms of all circadian genes, but usually caused disruptions to circadian rhythms to certain degrees (J Pharm Sci. 2017;106(9):2491-2498// Sci Transl Med. 2016;8(324):324ra16 // Mol Metab. 2013;2(3):292-305. // Nature. 2012;485(7396):123-7). This is most likely because circadian clock (biological clock) is a complicated system whose functions can not be determined by only one of its components." While disrupting most single core clock genes does not completely abolish rhythms, genetic deletion or suppression of Bmal1 does in fact abolish the vast majority of transcriptional rhythms. Kornmann et al PLoS Biology (2007) is the best evidence for this. Specifically as mentioned in my initial review, deletion of Bmal1 only in hepatocytes completely abolishes rhythmic Por expression. Although the authors state that they have re-performed this experiment and that their new data "show decreased Por levels and blunted rhythms consistent with the Lamia paper", that is not the case. They still see very robust rhythmic expression of Por which is absolutely not consistent with prior literature including the lamia 2008 paper and the Kornmann 2007 paper.

Response: We have gone back to original data, and identified some mistakes in data analyses of Por expression. We regret for this careless. The data have been fixed (please new Supplementary Figure 13). Por rhythmic expression is indeed abolished in the liver of Bmal1^{-/-} mice consistent with prior literature.

4. Some additional data presented conflict with prior literature and are difficult to understand. For example, the timing of rhythmic expression for many of the transcripts in this manuscript differ significantly from their expression patterns in other papers. See Koike et. al. (2012) supplementary data sets in table S7 where the peak of Dbp expression is ~CT8 rather than ~CT14 here.

I apologize I may have misread the peak time of Dbp in this manuscript.

5. Koike et al. identified a chromatin-binding site for endogenous BMAL1 at ~-1.5kb from the TSS in the Cyp3a11 locus in mouse livers. The experiment shown in Figure 3B does not exclude the possibility that this is an important site for direct activation of Cyp3a11 expression by BMAL1. The only expression construct that abolishes activation by BMAL1 is the one with no promoter at all so if any of the elements are present and conditions are saturating, it is not possible to identify specific active elements. To demonstrate that only the DR1 and D-box elements are required for activation by BMAL1, the experiment shown in Fig 4G needs to be repeated with the mutations introduced into the full length promoter (-2.0kb) element. None of the data exclude the possibility that BMAL1 acts both directly and indirectly to regulate Cyp3a11. A similar issue is also relevant for Supplementary Figure 6 regarding the HNF4a responsive elements.

6. Figures 4D/E are making the point that BMAL1 is being recruited to the Hnf4a promoter, however, Figure 4D only shows that some nuclear protein in the context of BMAL1 overexpression is binding the promoter elements, not necessarily BMAL1 itself. While 4E does demonstrate that lack of BMAL1 overexpression reduces binding of this protein to the promoter, this could also be explained by a protein induced by BMAL1 binding this region; overall, these experiments could be strengthened by performing this assay in cells lacking BMAL1 (both nontransfected and overexpressing BMAL1)

Okay.

7. In Fig 4F, the divided y-axis obscures the first column of data – it is unclear how high is the control IgG signal for the Dbp-Ebox.

Okay.

Minor Issues: ALL OK.

1. Several spelling/grammatical errors: Page 4 (CLCOK should be CLOCK); Page 5 (“robustness of circadian circuit” should be “circuits”); Page 15 (“PCR genotyping and expressing profiling” should be “expression”). The manuscript would benefit from additional editing for language and grammar.

2. Page 17 cites ChIP experiments showing Bmal1 recruitment to the E-Box of Dbp as

Figure 4E, but this is not the correct figure.

3. Figure 4G is not cited in the text.

4. Some figures are cited out of order, including Supplementary Fig 5.

5. It has been demonstrated that CRY repressors affect expression of Cyp3a11 in mouse livers, which should be cited. (Kriebs et al PNAS 2017).

Response: The reviewer is OK with all these previous responses. No further responses are needed.

REVIEWERS' COMMENTS:

Reviewer #1 (Remarks to the Author):

My comments have been handled properly well.

Regarding Table S6, the value of the mesors should be given as well, and the 95% confidence limits for meson, amplitude and acrophase should be provided. Regarding acrophase timing, it should be said what is the reference time "0". Expectedly, ZT0.

The p-value from the rejection of the non-null amplitude hypothesis for a 24-h period rhythm should be provided for each variable tested. If a comparison was made between Bmal1 proficient and Bmal1^{-/-} mice, it should be stated whether it involves a comparison of the mesons or the amplitude-phase vectors by Binham test.

Reviewer #2 (Remarks to the Author):

I am satisfied with the authors' responses. I would recommend changing "genetic mice" to "genetically-modified mice".

Reviewer #3 (Remarks to the Author):

Okay.

Response to the reviewer's comments

Reviewer #1

My comments have been handled properly well.

Regarding Table S6, the value of the mesors should be given as well, and the 95% confidence limits for mesor, amplitude and acrophase should be provided.

Response: Mesors have been added to Supplementary Tables 6 & 7. Additionally, we have provided standard errors (SE) for the estimated parameters (mesor, amplitude and acrophase) from Cosinor analyses.

Regarding acrophase timing, it should be said what is the reference time "0". Expectedly, ZT0.

Response: The reference time ZT17 (=12 PM) has been specified in the titles of Supplementary Tables 6 & 7.

The p-value from the rejection of the non-null amplitude hypothesis for a 24-h period rhythm should be provided for each variable tested.

Response: The p values for Cosinor analyses have been provided in the revised version (please see Supplementary Tables 6 & 7).

If a comparison was made between Bmal1 proficient and Bmal1^{-/-} mice, it should be stated whether it involves a comparison of the mesors or the

amplitude-phase vectors by Binham test.

Response: There may be a misunderstanding here. In fact, the comparisons between two genotypes were made based on two-way ANOVA with a post hoc test. The two-way ANOVA (F test) was able to identify whether the effect of genotype on gene expression was significant or not.

Reviewer #2

I am satisfied with the authors' responses. I would recommend changing "genetic mice" to "genetically-modified mice".

Response: Revised as suggested.

Reviewer #3

Okay.

Response: N/A.